# Phonon promoted charge density wave in topological kagome metal ScV$_6$Sn$_6$

Yong Hu [1,2,21] ✉, Junzhang Ma [3,4,5,21], Yinxiang Li [6,21], Yuxiao Jiang [7], Dariusz Jakub Gawryluk [8], Tianchen Hu [9], Jérémie Teyssier [10], Volodymyr Multian [11], Zhouyi Yin [12], Shuxiang Xu [9], Soohyeon Shin [8], Igor Plokhikh [8], Xinloong Han [13], Nicholas C. Plumb [1], Yang Liu [14], Jia-Xin Yin [15], Zurab Guguchia [16], Yue Zhao [12], Andreas P. Schnyder [17], Xianxin Wu [18] ✉, Ekaterina Pomjakushina [8], M. Zahid Hasan [7], Nanlin Wang [9,19,20] & Ming Shi [1,14] ✉

Charge density wave (CDW) orders in vanadium-based kagome metals have recently received tremendous attention, yet their origin remains a topic of debate. The discovery of ScV$_6$Sn$_6$, a bilayer kagome metal featuring an intriguing $\sqrt{3} \times \sqrt{3} \times 3$ CDW order, offers a novel platform to explore the underlying mechanism behind the unconventional CDW. Here, we combine high-resolution angle-resolved photoemission spectroscopy, Raman scattering and density functional theory to investigate the electronic structure and phonon modes of ScV$_6$Sn$_6$. We identify topologically nontrivial surface states and multiple van Hove singularities (VHSs) in the vicinity of the Fermi level, with one VHS aligning with the in-plane component of the CDW vector near the $\bar{K}$ point. Additionally, Raman measurements indicate a strong electron-phonon coupling, as evidenced by a two-phonon mode and new emergent modes. Our findings highlight the fundamental role of lattice degrees of freedom in promoting the CDW in ScV$_6$Sn$_6$.

Exploring exotic electronic orders and their underlying driving forces is a central issue in the field of quantum materials. One prime example is the emergence of charge density wave (CDW) order in cuprates, which is closely tied to unconventional superconductivity and spin density waves[1]. While cuprates exhibit strong electronic correlation effects, electron-boson coupling (e.g., electron-phonon coupling) is considered to be an indispensable contributor to both CDW and superconductivity[2,3]. Recent experimental efforts on kagome metals have also shed light on a potential connection between electron-phonon coupling and CDW formation.

The kagome lattice, a corner-sharing triangle network, has emerged as a versatile platform for exploring unconventional correlated and topological quantum states. Due to the unique correlation effects and frustrated lattice geometry inherent to kagome lattices, several families of kagome metals have been found to display a variety of competing electronic instabilities and nontrivial topologies, including quantum spin liquid[4–6], unconventional

superconductivity[7–10], charge density wave (CDW) orders[8–10], and Dirac/Weyl semimetals[11–13]. Of particular interest are the recently discovered non-magnetic vanadium-based superconductors $A$V$_3$Sb$_5$ ($A$ = K, Rb, Cs), which exhibit intriguing similarities to correlated electronic phenomena observed in high-temperature superconductors, such as CDW[14–16], pair density wave[17], and electronic nematicity[18]. Especially, the three-dimensional (3D) CDW order with an in-plane 2 × 2 reconstruction possesses exotic properties, including time-reversal symmetry breaking[15,19,20], intertwined with unconventional superconductivity[20], and rotational symmetry breaking[16,17]. Two possible scenarios, namely phonon softening[21,22] and correlation-driven Fermi surface (FS) instability[23–26], have been proposed to account for the CDW order. However, despite intense research efforts, the origin of the CDW order and its symmetry-breaking characteristics remain elusive.

Very recently, a new family of vanadium-based bilayer kagome metals, $R$V$_6$Sn$_6$ (where $R$ represents a rare-earth element), has been discovered[27,28]. Although its kagome layer does not show long-range

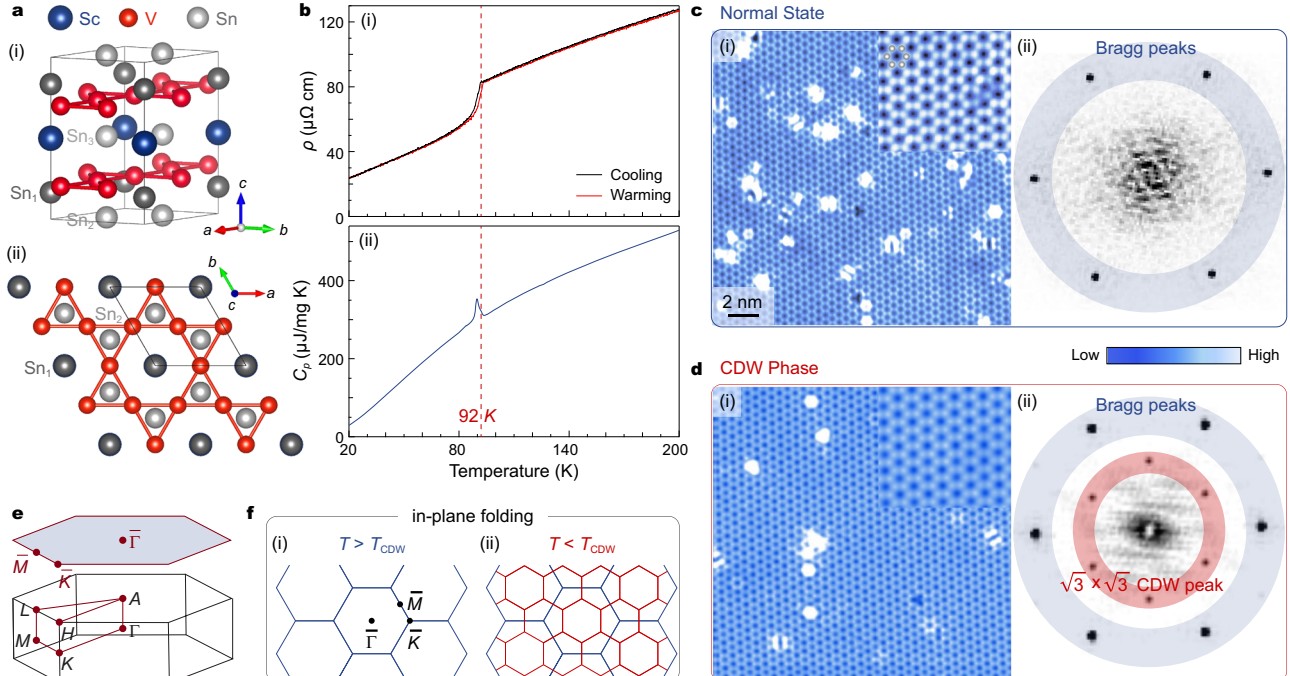

**Fig. 1 | Crystal structure, transport and topographic characterizations of ScV$_6$Sn$_6$. a** Crystal structure in the normal state showing the unit cell (i) and top view displaying the kagome lattice (ii). **b** Temperature-dependent *ab*-plane resistivity (i) and specific heat capacity (ii) of ScV$_6$Sn$_6$, indicating the onset of CDW near 92 K. **c** STM topograph of Sn$^2$ termination measured on the sample with $T_{CDW}$ = 80 K at 80 K (i) and associated Fourier transform (ii). Atomic Bragg peaks are highlighted with a blue ring. **d** Same data as in **c**, but taken at 1 K. $\sqrt{3} \times \sqrt{3}$ *R*30° CDW peaks are marked with a red ring. **e** Schematic of the bulk and surface Brillouin zones (BZs), with high-symmetry points marked. **f** Schematics of the in-plane folding of the surface BZ. Pristine (i) and CDW (ii) BZs are shown with blue and red lines, respectively.

magnetic order, similar to $A$V$_3$Sb$_5$, magnetism can be introduced by controlling the $R$ sites, providing a tunable platform to investigate magnetism, nontrivial topology and correlation effects native to the kagome lattice. Notably, ScV$_6$Sn$_6$, a member of the bilayer kagome family, undergoes an intriguing 3D CDW phase transition with a wavevector **Q** = (1/3, 1/3, 1/3) below $T_{CDW}$ ~ 92 K[29], unlike the CDW observed in $A$V$_3$Sb$_5$. Interestingly, recent experimental evidence shows that time-reversal symmetry breaking also occurs in the CDW state[30]. However, the nature of the CDW and its driving force remain unresolved. An in-depth investigation of the band structure and its interplay with lattice vibrations in ScV$_6$Sn$_6$ would provide valuable insights into the mechanism underlying the CDW order with intriguing symmetry-breaking in kagome metals.

Here, we investigate the electronic and lattice degrees of freedom in the CDW formation of the kagome metal ScV$_6$Sn$_6$ using a combination of scanning tunneling microscopy (STM), high-resolution angle-resolved photoemission spectroscopy (ARPES), Raman scattering measurements and density functional theory (DFT). Our low-temperature STM topographs visualize an in-plane $\sqrt{3} \times \sqrt{3}$ *R*30° reconstruction, corresponding to the bulk CDW wavevector measured in diffraction experiments[29]. In the electronic structure, we identify topologically nontrivial Dirac surface states (TDSSs) and multiple van Hove singularities (VHSs) in the vicinity of the Fermi level ($E_F$). Intriguingly, the nesting vector connecting the VHSs near the $\bar{K}$ point is close to (1/3, 1/3), matching with the observed $\sqrt{3} \times \sqrt{3}$ *R*30° CDW wave vector. In contrast to $A$V$_3$Sb$_5$, however, pronounced band reconstructions appear to be absent in the CDW state of ScV$_6$Sn$_6$, possibly due to the 3D nature of the $\sqrt{3} \times \sqrt{3} \times 3$ CDW order and a noticeable dispersion along the *c*-direction. Remarkably, our Raman measurements reveal the presence of a two-phonon mode in the normal state and new emergent Raman-active phonon modes in the CDW phase, indicating a strong electron-phonon coupling. Collectively, our results emphasize the crucial role of lattice degrees of freedom in promoting the CDW in ScV$_6$Sn$_6$ and contribute to a deeper understanding of the diverse quantum correlation phenomena observed in vanadium-based kagome metals.

## Results

### Superlattice modulation of the CDW visualized by STM

The pristine phase of ScV$_6$Sn$_6$ crystallizes in a layered structure with the space group *P*6/*mmm*. The unit cell consists of two V$_3$Sn$^1$ kagome layers, with Sn$^2$ and ScSn$^3_2$ layers stacked in an alternating fashion along the out-of-plane direction (*c*-axis) (Fig. 1a). Similar to the sister compound GdV$_6$Sn$_6$, ScV$_6$Sn$_6$ tends to cleave along the *c*-axis, resulting in three surface terminations (see Supplementary Note 1 for details), namely the kagome (V$_3$Sn), ScSn$^3_2$ and Sn layers[31,32]. The electrical resistivity [Fig. 1b(i)] and specific heat capacity [Fig. 1b(ii)] measurements consistently show a transition around 92 K, indicating the presence of the CDW transition[29,33]. To study the superlattice modulation of the CDW, we perform comparative STM measurements on ScV$_6$Sn$_6$ in both the normal state (Fig. 1c) and the CDW phase (Fig. 1d). Atomically resolved STM topographies clearly identify the hexagonal lattice formed by the Sn$^2$ atoms [Fig. 1c(i) and 1d(i)] and the in-plane modulation in the CDW phase [inset of Fig. 1d(i)]. The Fourier transform of the topographic data further visualizes the existence of a $\sqrt{3} \times \sqrt{3}$ *R*30° reconstruction in the CDW phase [Fig. 1d(ii)], which is absent in the normal state [Fig. 1c(ii)]. Figure 1e displays the bulk Brillouin zone (BZ) and the projected two-dimensional surface BZ. In the CDW phase, the in-plane component of the CDW folds the pristine BZ [Fig. 1f(i)] into the new smaller $\sqrt{3} \times \sqrt{3}$ BZ [Fig. 1f(ii)].

### $\mathbb{Z}_2$ topological surfaces states, Dirac cones and VHSs identified by ARPES

We next focus on the electronic structure of the bilayer kagome metal ScV$_6$Sn$_6$ (Fig. 2). Utilizing high-resolution ARPES measurements with a small beam spot, we reveal three different sets of ARPES spectra

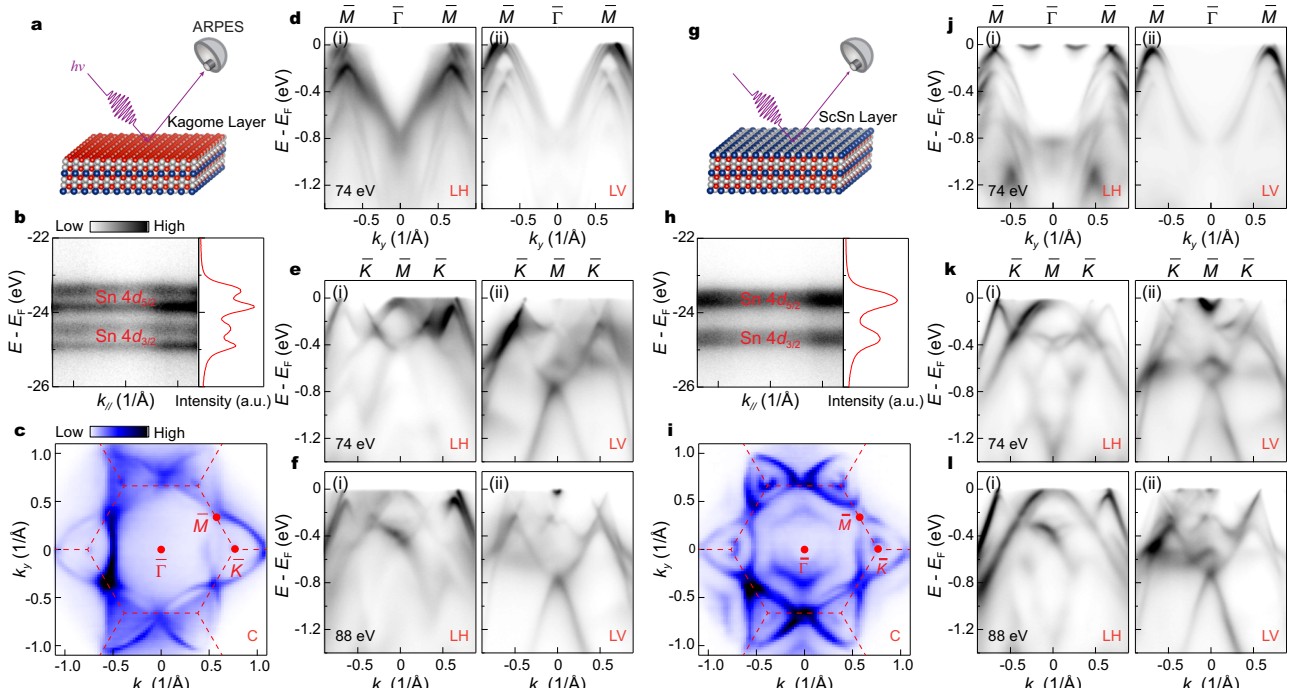

**Fig. 2 | Termination-dependent photoemission measurements of the electronic structure in ScV$_6$Sn$_6$. a** Schematic of the kagome termination. **b** The corresponding x-ray photoelectron spectroscopy (XPS) spectrum on the Sn $4d$ core levels (left) and the integrated energy distribution curve (EDC) of the core levels (right). **c** Fermi surface (FS) mapping collected on the kagome termination, measured with circularly (C) polarized light. The dashed red line represents the pristine

BZ. **d** Photoelectron intensity plots of the band structure taken along the $\bar{\Gamma}$ - $\bar{M}$ direction on the kagome termination, measured with 74 eV linear horizontal (LH) (i) and linear vertical (LV) (ii) polarizations. **e**, **f** Same data as in **d**, but taken along the $\bar{\Gamma}$-$\bar{K}$ direction, measured with 74 eV (**e**) and 88 eV (**f**). **g**–**l** Same data as in **a**–**f**, but measured on the ScSn$_2^3$ termination.

associated with the possible surface terminations on the cleaved sample surface (Fig. 2a, g and Supplementary Fig. 1). As previously established in GdV$_6$Sn$_6$ compound[32], the surface terminations of the sample can be identified by measuring the Sn $4d$ core level (Fig. 2b, h). In Fig. 2b–f, h–l, we present the electronic structure from the kagome (Fig. 2a) and ScSn$_2^3$ terminations (Fig. 2g), respectively. The measured FS, with pristine BZ (the dashed red lines in Fig. 2c, i) and high-symmetry points ($\bar{\Gamma}$, $\bar{K}$ and $\bar{M}$) labeled, features a characteristic hexagonal kagome Fermiology, as generally exhibited in other kagome systems[11,13,14,29,34]. To visualize the energy-momentum dispersion of the electronic structure, polarization-dependent measurements are performed along two different high-symmetry paths, $\bar{\Gamma}$ - $\bar{M}$ (Fig. 2d, j) and $\bar{\Gamma}$ - $\bar{K}$ (Fig. 2e, k) directions. Similar to other vanadium-based kagome metals[34,35], the photoemission intensities are strongly sensitive to the photon polarization (Fig. 2d–f, j–l), reflecting the multi-orbital nature of V-$d$ orbitals. In contrast to $A$V$_3$Sb$_5$, the photon-energy dependent measurements along the $\bar{\Gamma}$ - $\bar{K}$ direction (Fig. 2e, f, k, l) exhibit distinct band dispersions at different $k_z$ planes (also see Supplementary Fig. 2), indicating the relatively obvious three-dimensionality of the electronic structure in ScV$_6$Sn$_6$.

A comparative examination of the band structure on different terminations reveals some differences, which can be attributed to the presence of surface states and matrix element effects. Specifically, on the ScSn$_2^3$ termination, the measured FS centered around the $\bar{\Gamma}$ point consists of a circular-shaped and a hexagonal-shaped FS sheet, as illustrated in Fig. 3a. A closer inspection of the band dispersion, as depicted in Fig. 3b, indicates that these FS sheets around $\bar{\Gamma}$ arise from two V-shaped bands, which bear a striking resemblance to the TDSSs previously observed in GdV$_6$Sn$_6$[32]. Our DFT calculations (see Supplementary Fig. 3 for details) confirm these observations and highlight that the observed V-shaped bands around the $\bar{\Gamma}$ point (Fig. 3b) are indicative of the existence of TDSSs originating from a $\mathbb{Z}_2$ bulk topology in ScV$_6$Sn$_6$.

Furthermore, in addition to the TDSSs, the ARPES spectra collected on the ScSn$_2^3$ termination reveal more details of the kagome bands, such as the characteristic Dirac cone (DC) and VHSs expected from the kagome tight-binding model[8,9]. Constant energy maps shown in Fig. 3c reveal two Dirac cones around the $\bar{K}$ point. The energy–momentum dispersion along the $\bar{\Gamma}$ - $\bar{K}$ - $\bar{M}$ - $\bar{\Gamma}$ direction (Fig. 3d), which agrees well with the calculated bulk states projected onto the (001) surface (Fig. 3e), confirms the existence of Dirac cones at binding energies ($E_B$) of 0.09 eV (DC1) and 0.28 eV (DC2), despite some differences in the energy positions of Dirac points between experimental data and theoretical calculations. Additionally, the band forming the DC2 extends to the $\bar{M}$ point and constitutes a VHS (labeled as VHS1 in Fig. 3d, e). The saddle point nature of VHS1 is evident from cuts taken vertically across the $\bar{K}$ - $\bar{M}$ path (#M$_1$-#M$_5$, as indicated in Fig. 3f), where the band bottom of the electron-like band (dashed green curve in Fig. 3g) exhibits a maximum energy slightly above $E_F$ at the $\bar{M}$ point (green solid curve). Furthermore, another hole-like band observed in Fig. 3h (same as Fig. 3g), which is slightly below the VHS1 band, has a minimum energy at the $\bar{M}$ point, indicating the electron-like nature along the orthogonal direction (the blue curve in Fig. 3h). This feature demonstrates another van Hove band with the opposite dispersion close to $E_F$ (marked as VHS2 in Fig. 3d). These twofold concavity VHSs are consistent with theoretical calculations (Fig. 3e) and have been identified in $A$V$_3$Sb$_5$[34,36], where they are believed to promote the CDW order. Interestingly, we also identify an unusual VHS (referred to as VHS3) contributed by the DC1 band near the $\bar{K}$ point, as highlighted by the dashed red curve in Fig. 3d. To further confirm the saddle point nature of VHS3, we examine the band dispersions perpendicular to the $\bar{\Gamma}$ - $\bar{K}$ direction (cuts #K$_1$-#K$_5$ in Fig. 3f, i). The series of cuts in Fig. 3i reveal a hole-like band (dashed red curve) with a band bottom that exhibits a minimum energy at the $\bar{\Lambda}$ point. These are fully consistent with our calculations shown in the inset of Fig. 3e, confirming its van Hove nature (see also Supplementary Fig. 4). Due to the

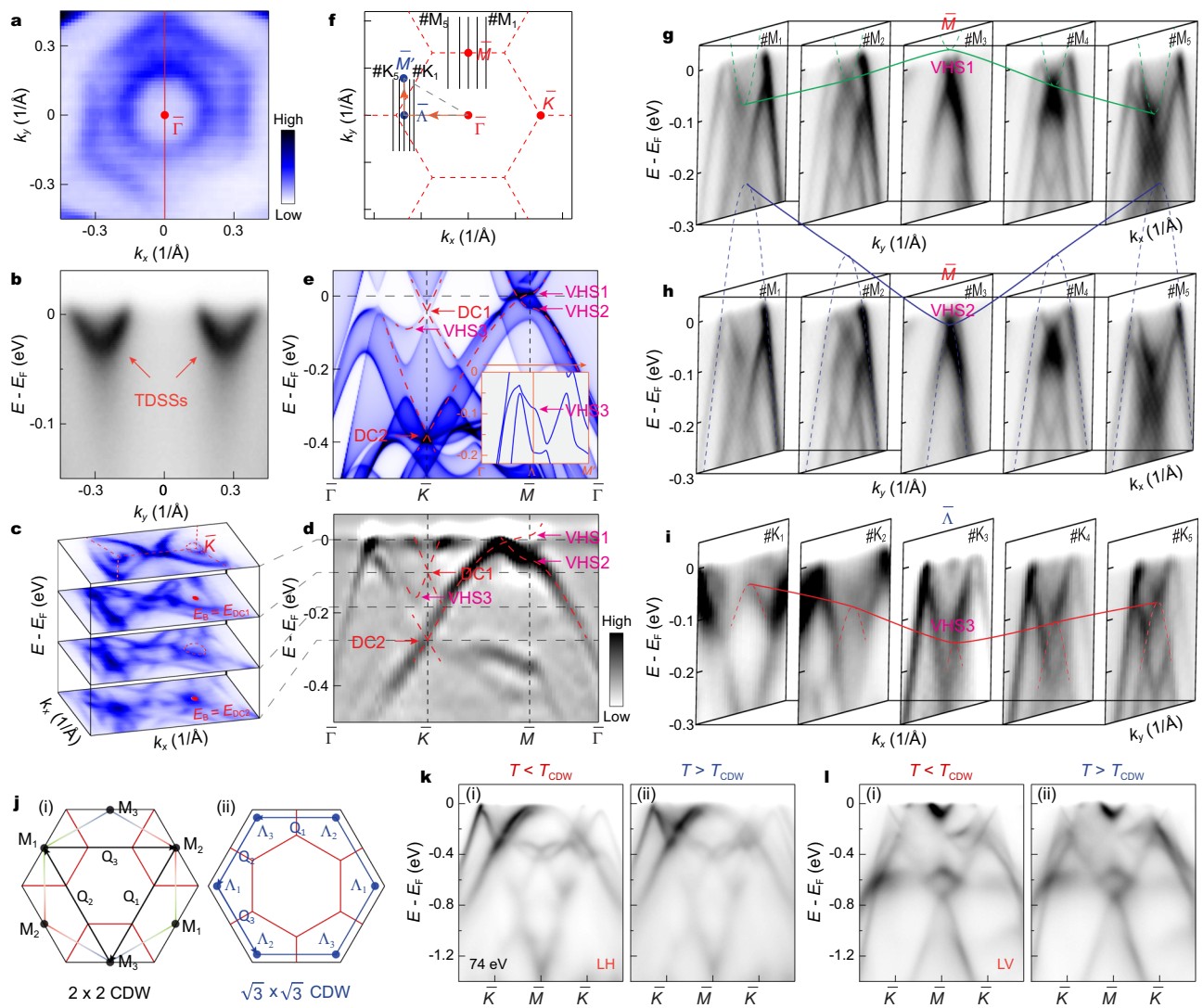

**Fig. 3 | $\mathbb{Z}_2$ topological surfaces states, Dirac cones and VHSs in ScV$_6$Sn$_6$. a** Zoom-in FS mapping measured on the ScSn$^3_2$ termination. **b** ARPES spectrum taken along the $\bar{\Gamma}$ - $\bar{M}$ direction highlighting the TDSSs. The momentum path is indicated by the red line in **a**. **c** Stacking plots of constant energy maps around the $\bar{K}$ point. **d** Second derivative spectrum as a function of along the $\bar{\Gamma}$ - $\bar{K}$ - $\bar{M}$ - $\bar{\Gamma}$ direction. **e** Calculated band structure along the $\bar{\Gamma}$ - $\bar{K}$ - $\bar{M}$ - $\bar{\Gamma}$ direction. Dirac cone (DC) and VHS are indicated by red and pink arrows, respectively. The inset displays DFT bands along the $\Gamma$ - $\Lambda$ - $M'$ direction (as indicated by the orange arrow in **f**). **f** Schematics of the surface BZ. **g** A series of cuts taken vertically across the $\bar{K}$ - $\bar{M}$ path, the momentum paths of the cuts (#M$_1$-#M$_5$) are indicated by the black lines in **f**. Dashed green curve highlights the electron-like band and solid curve indicates the corresponding VHS1

at the $\bar{M}$ point. **h** Same data as in **g**, but highlights the hole-like band (dashed blue curve) and VHS2 (solid curve). **i** Stack of cuts perpendicular to the $\bar{\Gamma}$ - $\bar{K}$ direction. The momentum directions of the cuts (#K$_1$-#K$_5$) are indicated by the black lines in **f**. Dashed red curve and solid curve indicate the hole-like band and corresponding VHS3 at the $\bar{\Lambda}$ point, respectively. The ARPES spectra shown in **g**–**i** were taken with 74 eV C polarized light. **j** FS of kagome lattice at the VHS filling. The three inequivalent saddle points M$_i$ ($\Lambda_i$) are connected by three inequivalent nesting vectors Q$_i$, which can give rise to a 2 × 2 CDW (i) and $\sqrt{3} \times \sqrt{3}$ CDW (ii). **k** Temperature-dependent measurements of the band structure along the $\bar{\Gamma}$ - $\bar{K}$ direction, measured below $T_{CDW}$ at 20 K (i) and above $T_{CDW}$ at 130 K (ii) with 74 eV LH polarized light. **l** Same data as in **b**, but measured with LV polarization.

six-fold rotational symmetry of the lattice, there are six such saddle points near $\bar{K}$ and $\bar{\Lambda}$, as shown in Fig. 3j(ii).

As VHSs carry large density of states and can promote competing electronic instabilities, we now explore the potential contribution of the identified multiple VHSs to the CDW in ScV$_6$Sn$_6$. Previous theoretical studies have emphasized that VHSs located at the M point can naturally give rise to nesting vectors Q$_{1,2,3}$[8,9] that connect different sublattices on the saddle points of the FS [Fig. 3j(i)], potentially leading to a 2 × 2 bond CDW instability. However, we note that the suggested FS nesting wave vectors Q$_{1,2,3}$ in Fig. 3j(i) are incompatible with the in-plane $\sqrt{3} \times \sqrt{3}$ $R30°$ reconstruction observed in ScV$_6$Sn$_6$ [Fig. 1d(ii)]. Nevertheless, the nesting vectors associated with the identified VHS3 near the $\bar{K}$ point are in proximity to (1/3, 1/3) [Fig. 3j(ii)], which is more consistent with the observed in-plane $\sqrt{3} \times \sqrt{3}$ CDW pattern. To assess

the role of VHS3 in the CDW formation, we perform temperature-dependent measurements on the band dispersions along the $\bar{\Gamma}$ - $\bar{K}$ direction. Surprisingly, our high-resolution ARPES spectra show negligible differences between the CDW phase and the normal state (Fig. 3k, l), in contrast to the significant band reconstructions observed in $A$V$_3$Sb$_5$[35,37,38]. As the V-3$d$ states dominate near the $E_F$ (Supplementary Fig. 5), this weak band reconstruction and folding effect may be due to the 3D nature of the $\sqrt{3} \times \sqrt{3} \times 3$ CDW order and the noticeable dispersion along the $c$-direction (Fig. 2e, f, k, l and Supplementary Fig. 2).

**Two-phonon mode revealed by Raman**

After investigating the electronic structure of ScV$_6$Sn$_6$, we next examine the effects of the lattice degrees of freedom on the CDW

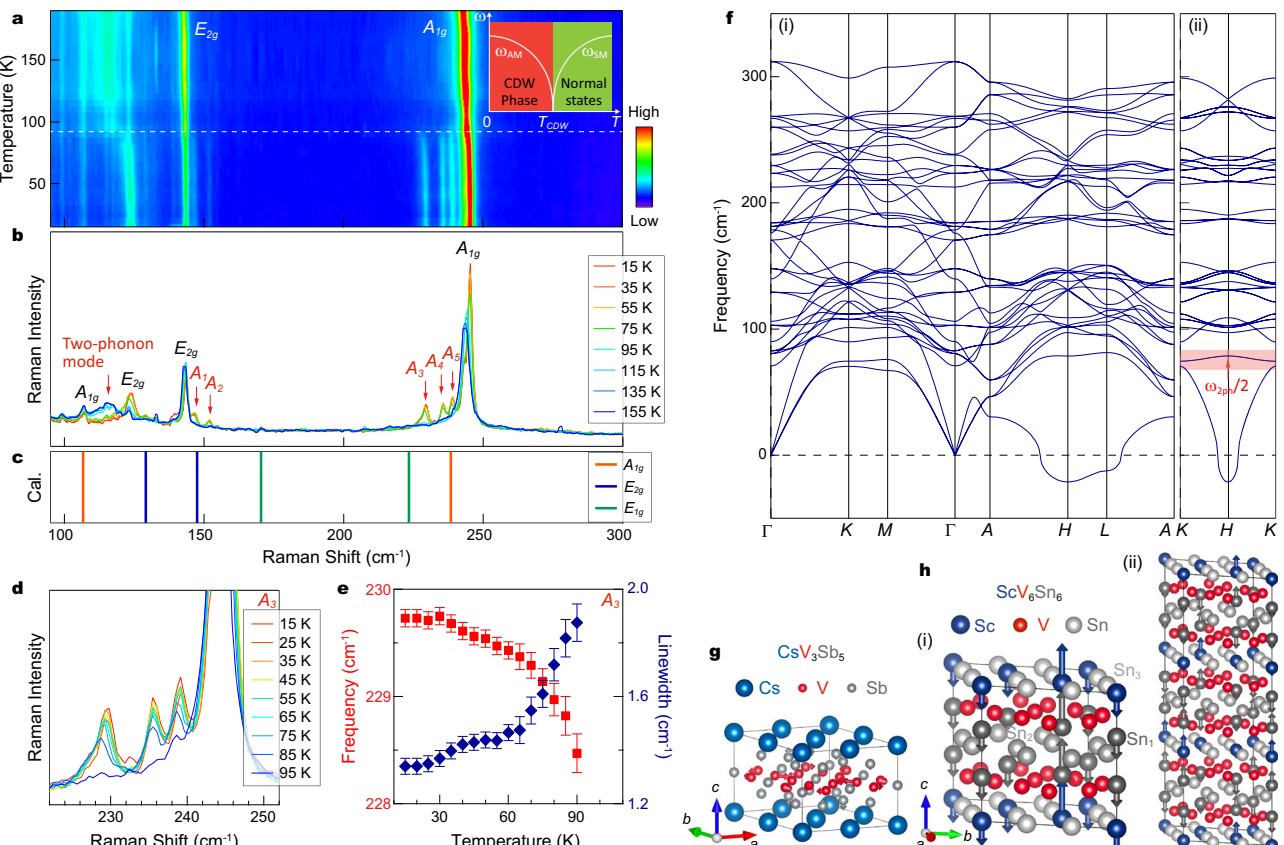

**Fig. 4 | Raman modes and phonon band structure in ScV$_6$Sn$_6$. a** Temperature-dependent colormap of the Raman response recorded on ScV$_6$Sn$_6$. The inset illustrates the relationship between the soft mode and amplitude mode in typical CDW materials. The soft mode frequency ($\omega_{SM}$) freezes below $T_{CDW}$, and the frequency ($\omega_{AM}$) emerges afterward. **b** Typical Raman spectra measured below and above $T_{CDW}$. **c** Calculated Raman mode frequencies around $\Gamma$ point. **d** Temperature-dependent Raman spectra illustrating the $A_3$ mode. **e** Temperature dependence of the Frequency and linewidth of the $A_3$ mode. The error bars represent the fit uncertainty. **f** DFT calculated phonon band structure along high-symmetry paths (i) and the $K$ - $H$ - $K$ path (ii) of pristine ScV$_6$Sn$_6$, with experimental lattice parameters[29]. The red shaded region indicates the half frequency of the two-phonon mode. **g** Distortion pattern of the trihexagonal pattern in the 2 × 2 × 1 CDW phase of CsV$_3$Sb$_5$. **h** Acoustic phonon mode at the $K$ point (i) corresponding to the observed two-phonon mode in **a**, or **b**; and distortion pattern of the $\sqrt{3} \times \sqrt{3} \times 3$ CDW (ii) indicated by the vectors, with respect to the pristine phase of ScV$_6$Sn$_6$ [Fig. 1a(i)]. The length of the vectors represents the amplitude of atomic displacements.

formation using Raman scattering. Figure 4a displays a colormap of the Raman response, covering a temperature from 15 K to 190 K. The Raman spectra (Fig. 4a, b) feature two prominent Raman-active phonon peaks at 143 cm$^{-1}$ and 243 cm$^{-1}$, which we attribute to the $E_{2g}$ and $A_{1g}$ modes, respectively, based on the polarization-dependent measurements (for details see Supplementary Fig. 6) and theoretical calculations (Fig. 4c). Additionally, we observe a broad peak around 116 cm$^{-1}$ (highlighted by the dashed white curve in Fig. 4a and red arrow in Fig. 4b) in the spectra above $T_{CDW}$ (indicated by the dashed white line in Fig. 4a). As the temperature decreases, both the $E_{2g}$ and $A_{1g}$ modes exhibit a blueshift (Supplementary Fig. 6), while the broad peak around 116 cm$^{-1}$ shifts minimally above $T_{CDW}$, but abruptly vanishes below $T_{CDW}$ (Fig. 4b). The temperature-dependent behavior of the broad peak resembles the one observed in other well-studied CDW materials[39–42], indicating the presence of a two-phonon Raman mode. This mode involves two phonons with opposite wave vectors and represents a second order process usually correlated with the strong momentum dependent electron-phonon coupling near the CDW wave vector[40,43–45]. In ScV$_6$Sn$_6$, the observed two-phonon mode likely originates from the acoustic longitudinal modes in the $K$-$H$ path [Fig. 4f(ii), the shaded region highlights the half frequency of the two-phonon mode], according to the theoretical phonon dispersion in Fig. 4f. Below $T_{CDW}$, the two-phonon mode disappears, possibly due to CDW-induced phonon folding and alteration of electron-phonon coupling. Moreover, multiple new phonon peaks (labeled as $A_1$-$A_4$ in Fig. 4b)

emerge below $T_{CDW}$, around 150 cm$^{-1}$ and 240 cm$^{-1}$, indicating their intimate relationship with the CDW order. Interestingly, the two new modes ($A_1$, $A_2$) close to the $E_{2g}$ have almost no specific temperature dependence in their frequencies and linewidths (Fig. 4a, b) as the temperature approaches $T_{CDW}$, consistent with characteristics of CDW zone-folded modes. In contrast, the $A_3$ mode shows noticeable softening and broadening with warming towards $T_{CDW}$ (Fig. 4d, e), eventually becoming unresolvable above $T_{CDW}$ (Fig. 4a, b, d, e). These observations are potentially indicative of a CDW amplitude mode (see Supplementary Note 6 for details) derived from the collapse of coherent CDW order near $T_{CDW}$[39–42,46].

Our theoretical calculations show that imaginary phonon modes appear at the $H$ and $L$ points (Fig. 4f), corresponding to $\sqrt{3} \times \sqrt{3} \times 2$ and 2 × 2 lattice reconstructions, respectively. However, these modes, along with the absence of unstable phonon modes at (1/3, 1/3, 1/3) (Fig. 4f), fail to explain the observed $\sqrt{3} \times \sqrt{3} \times 3$ CDW order. This suggests that the bare phonon instability is insufficient to account for the CDW order in our experiments (Fig. 1b, d). Our identification of the two-phonon mode order, typically much weaker than the one-phonon Raman modes, points to a strong electron-phonon coupling in ScV$_6$Sn$_6$. This coupling could induce significant phonon softening at the CDW vector by introducing a negative self-energy term through the electron bubble[43]. Consequently, the renormalized phonon dispersion may exhibit an anomaly and a minimum negative frequency at the wave vector $\mathbf{Q} = $ (1/3, 1/3, 1/3), giving rise to the observed CDW

order. Our Raman measurements support this scenario, as they show the absence of one-phonon softening modes and the observation of new emergent phonon modes with high frequency[47].

## Discussion

Finally, we delve into the potential origin of the CDW order in the kagome metal $ScV_6Sn_6$ based on our combined ARPES and Raman measurements. From an electronic structure perspective, our ARPES measurements unveil multiorbital characteristics and a relatively apparent three-dimensionality in the electronic states of $ScV_6Sn_6$. Interestingly, the temperature dependence of the electronic structure shows negligible differences between the V-kagome bands in the CDW phase and the normal state. These findings suggest a weak band reconstruction and folding effect in $ScV_6Sn_6$, indicating a small distortion of the V atoms. Therefore, our ARPES measurements imply that Fermi surface nesting may not be the primary driving force behind the CDW in $ScV_6Sn_6$. However, the nesting vector between the observed VHSs around the $\bar{K}$ point [Fig. 3j(ii)] aligns with the in-plane component of the CDW wave vector, suggesting that electronic correlation might participate in promoting the in-plane component of CDW order[48].

The electronic landscape of $ScV_6Sn_6$ stands in sharp contrast to the observations in $AV_3Sb_5$, where the electronic structure is relatively two-dimensional and exhibits pronounced band reconstructions in its CDW state. These discrepancies in $AV_3Sb_5$ and $ScV_6Sn_6$ may arise from their distinct CDW patterns: In $AV_3Sb_5$, the CDW pattern mainly results from the distortion of the kagome V atoms (Fig. 4g)[21,35,49], while in $ScV_6Sn_6$, the CDW order mainly involves the displacement of Sc and Sn, while the V atoms show negligible distortion (Fig. 4h)[29].

From the perspective of lattice degrees of freedom, our Raman measurements reveal a two-phonon mode in the normal state and new emergent phonon modes with high frequency in the CDW phase, indicating a strong electron-phonon coupling. These observations imply that electron-phonon coupling may play a crucial role in promoting the CDW order in $ScV_6Sn_6$. Further considering the correlation effect associated with VHSs, we deduce that electron-phonon coupling and electron-electron interactions may conspire to generate the symmetry-breaking states in the vanadium-based kagome metals, which warrants further investigations, both from the theoretical and the experimental fronts.

In conclusion, our study combining ARPES and Raman scattering measurements provides important insights into the underlying mechanism of the CDW order in $ScV_6Sn_6$. The VHSs located near the $\bar{K}$ point could introduce nesting wave vectors close to (1/3, 1/3), which are consistent with the observed in-plane $\sqrt{3} \times \sqrt{3}$ CDW order. Furthermore, our Raman measurements demonstrate the presence of the two-phonon mode and new phonon modes, indicating a strong electron-phonon coupling[42]. Taken together, our results suggest a concerted mechanism of the CDW order in $ScV_6Sn_6$ involving both electron-phonon coupling and electron correlation effects. Further investigations are necessary to fully comprehend the interplay between these two mechanisms and their roles in promoting the unconventional CDW order in vanadium-based kagome metals.

## Methods

### Single crystals growth and characterization

This work studied two baches of $ScV_6Sn_6$ samples, obtained from two different research groups, with $T_{CDW} = 92$ K[33] and $T_{CDW} = 80$ K[30], respectively. Single crystals of $ScV_6Sn_6$ were synthesized using the Sn self-flux method.

For samples with $T_{CDW} = 92$ K, high-purity Sc, V, and Sn elements were mixed in a molar ratio of 1:6:40, placed in an alumina crucible, and sealed in a silica ampule under vacuum (see Ref. 33. for more details). The sealed ampule was then heated gradually to 1150 °C and held for 20 h before being cooled at a rate of 1 °C /h to 750 °C. At the final

temperature, the mixture was centrifuged, resulting in the formation of shiny hexagonal crystals. The temperature-dependent resistivity [Fig. 1b(i)] was measured using a four-probe method with the current perpendicular to the $c$-axis, while the heat capacity [Fig. 1b(ii)] was measured using the relaxation method, both of which were conducted in a Quantum Design physical property measurement system.

For samples with $T_{CDW} = 80$ K, the starting materials were Scandium (Alfa Aesar, 99.9%), vanadium (Alfa Aesar, 99.8%), and tin (Alfa Aesar, 99.9999%). The starting components were mixed in helium-filled glovebox. Mixture of Sc:V:Sn in molar ratio equal 1:6:58 and total weight 35.1 g was placed into a 5 ml alumina Canfield crucible. The crucible with material was inserted into a quartz ampule. The specimen was subsequently transferred to vacuum line, evacuated and sealed under ~100 mbar of backfilled argon. The sample was heated up to 1150 °C, with a rate 400 °C/h, and annealed at that temperature for 12 h. After isothermal step, the sample was cooled down to 780 °C with a rate of 1 °C/h. After that step, the excess of Sn flux was decanted from the single crystals by means of centrifugation (see Ref. 30. for more details). Electrical resistivity of single-crystal $ScV_6Sn_6$ was measured using the typical four-probe technique in the physical property measurement system (PPMS, QD). Four Pt-wires (one milli-inch diameter) were attached on the ab-plane of the bar-shaped specimen using silver epoxy.

### ARPES measurements

Angle-resolved photoemission spectroscopy (ARPES) measurements were carried out at the ULTRA endstation of the Surface/Interface Spectroscopy (SIS) beamline of the Swiss Light Source, using a Scienta Omicron DA30L analyzer. Energy and angular resolution were set to be around 15 meV and 0.1°, respectively. To reveal possible effect of the CDW on the band structure, ARPES measured the two baches of $ScV_6Sn_6$ samples with $T_{CDW} = 92$ K and $T_{CDW} = 80$ K. The quality of ARPES data from the two different $T_{CDW}$ samples is comparable and the corresponding results appear to be the same, *i.e.*, ARPES spectra show negligible differences between the CDW phase and the normal state. The samples were cleaved in-situ with a base pressure of better than $5 \times 10^{-11}$ torr, and (if not specified) measured at 20 K. The Fermi level was determined by measuring a polycrystalline Au in electrical contact with the samples.

### Raman measurements

Raman spectra were recorded on $ScV_6Sn_6$ samples with $T_{CDW} = 92$ K using a Horiba LabRAM HR Evolution spectrometer with an excitation wavelength of 532 *nm* and a resolution of 1 cm$^{-1}$ over the full range. The laser light was focused on a 2-μm spot using a window-corrected 63× objective. Samples were glued on the copper plate of a He flow cryostat (Konti Micro from CryoVac GMBH) using silver paint. Stokes and anti-stokes were simultaneously recorded and real temperature were check by fitting stokes and anti-stokes ratio using reffit software[50] (https://reffit.ch/).

### STM measurements

All measurements were performed in the UNISOKU USM-1500s system, using chemically etched tungsten tip. The sample is cleaved at liquid nitrogen temperature and immediately inserted into the SPM head in UHV environment. Junction set-ups are $V_{bias} = -300$ mV, $I = 200$ pA, and $V_{bias} = -200$ mV, $I = 100$ pA for 22 nm × 22 nm image and 6.8 nm × 6.8 nm atomic-resolution topography respectively. This topography indicates an $Sn^2$ cleavage plane. The lattice constant calculated by Fast Fourier Transform plot of the atomic resolution topography is 5.4 Å.

### Computational methods

Band structure calculations were performed by using the method of first-principle density functional theory (DFT) as implemented in

the Vienna ab initio simulation package (VASP) code[51–53]. The Perdew-Burke-Ernzerhof (PBE) exchange-correlation functional and the projector-augmented-wave (PAW) approach are used[54]. For the calculations of band structures, the cutoff energy is set to 500 eV for expanding the wave functions into plane-wave basis and the sample k-point mesh is $8 \times 8 \times 5$. Based on DFT band structures, the maximally localized Wannier functions (MLWFs) with 120 orbitals (Sc $s$, $d$; V $d$; Sn $s$, $p$;) were used to construct a tight-binding model[55]. Then, we use the Green's function of semi-infinite system for topological surface states calculation[56]. In the calculations of phonon dispersions, the finite displacement method implemented in the PHONOPY is used and the energy cutoff is 300 eV. The experimental lattice constants $a = b = 5.4669$ Å and $c = 9.1594$ Å[29] are adopted in our calculations.

## Data availability

All data needed to evaluate the conclusions in the paper are present in the paper and/or the Supplementary Information. All other data that support the findings of this study are available from the corresponding authors upon request. Source data (the.xls file for Fig. 1b and other relevant data) are available at a public repository (MARVEL Materials Cloud Archive) with the same title of this paper (https://archive.materialscloud.org).

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

## Acknowledgements

Y.H. was supported by the Swiss National Science Foundation under Grant. No. 200021_188413. J.Z.M. was supported by the Research Grants Council of Hong Kong via Early Career Scheme (21304023), the National Natural Science Foundation of China (12104379), Guangdong Basic and Applied Basic Research Foundation (2021B1515130007). X.W. is supported by the National Key R&D Program of China (Grant No.2023YFA1407300) and the National Natural Science Foundation of China (Grant No. 12047503). N.L.W is supported by National Natural Science Foundation of China (No. 11888101), the National Key R&D Program of China (No. 2022YFA1403901). M.S. and Y.L. were supported by the National Natural Science Foundation of China (12350710785). I.P. acknowledges support from Paul Scherrer Institute research grant No. 2021_01346. S.S. acknowledges support from the Swiss National Science Foundation (SNSF) (No. 200021_188706). J.X.Y. acknowledges the support from the National Key R&D Program of China (No. 2023YFA1407300) and the National Science Foundation of China (No. 12374060). Z.G. acknowledges support from the Swiss National Science Foundation (SNSF) through SNSF Starting Grant (No. TMSGI2_211750). A.P.S. thanks the YITP Kyoto for hospitality and the Deutsche Forschungsgemeinschaft (DFG, German Research Foundation) – TRR 360 – 492547816 for funding.

## Author contributions

Y.H. and M.S. conceived the ARPES experiments. D.J.G. grew and characterized the $T_{CDW}$ = 80 K crystals with the help from S.S., I.P. and the support from G.Z., E.P. T.H. grew and characterized the $T_{CDW}$ = 92 K crystals with the help from S.X. and the guidance from N.W.; X.W., Y.L. and X.H. performed the theoretical calculations and analysis, with the support from A.P.S.; Y.H. performed the ARPES experiments with the help from J.Z.M., Y.L. and M.S.; N.C.P. maintained the ARPES facilities at the SIS-ULTRA. Y.H. analyzed the ARPES data in discussion with J.Z.M. V.M. and Y.H. performed the Raman measurement, Y.H. analyzed the Raman data, with the guidance from J.T.; The STM experiments were performed by Y.J. and Z.Y. with guidance from Y.Z., J.-X.Y. and M.Z.H.; Y.H. wrote the paper with theoretical discussion with X.W.; M.S., Y.H. and X.W. supervised the project. All authors contributed to the discussion and comment on the paper.

## Competing interests

The authors declare no competing interests.

## Additional information

[1]Photon Science Division, Paul Scherrer Institut, CH-5232 Villigen PSI, Switzerland. [2]Center of Quantum Materials and Devices and Department of Applied Physics, Chongqing University, 401331 Chongqing, China. [3]Department of Physics, City University of Hong Kong, Kowloon, Hong Kong, China. [4]City University of Hong Kong Shenzhen Research Institute, Shenzhen, China. [5]Hong Kong Institute for Advanced Study, City University of Hong Kong, Kowloon, Hong Kong, China. [6]College of Science, University of Shanghai for Science and Technology, 200093 Shanghai, China. [7]Laboratory for Topological Quantum Matter and Advanced Spectroscopy (B7), Department of Physics, Princeton University, Princeton, NJ, USA. [8]Laboratory for Multiscale Materials Experiments, Paul Scherrer Institut, CH-5232 Villigen PSI, Switzerland. [9]International Center for Quantum Materials, School of Physics, Peking University, 100871 Beijing, China. [10]Department of Quantum Matter Physics, University of Geneva, 24 Quai Ernest-Ansermet, 1211 Geneva 4, Switzerland. [11]Advanced Materials Nonlinear Optical Diagnostics lab, Institute of Physics, NAS of Ukraine, 46 Nauky pr., 03028 Kyiv, Ukraine. [12]Institute for Quantum Science and Engineering and Department of Physics, Southern University of Science and Technology of China, Shenzhen, Guangdong 518055, China. [13]Kavli Institute for Theoretical Sciences, University of Chinese Academy of Sciences, 100190 Beijing, China. [14]Center for Correlated Matter and Department of Physics, Zhejiang University, 310058 Hangzhou, China. [15]Department of physics, Southern University of Science and Technology, 518055 Shenzhen, Guangdong, China. [16]Laboratory for Muon Spin Spectroscopy, Paul Scherrer Institute, CH-5232 Villigen PSI, Switzerland. [17]Max-Planck-Institut für Festkörperforschung, Heisenbergstrasse 1, D-70569 Stuttgart, Germany. [18]CAS Key Laboratory of Theoretical Physics, Institute of Theoretical Physics, Chinese Academy of Sciences, Beijing 100190, China. [19]Beijing Academy of Quantum Information Sciences, Beijing 100913, China. [20]Collaborative Innovation Center of Quantum Matter, Beijing 100871, China. [21]These authors contributed equally: Yong Hu, Junzhang Ma, Yinxiang Li. ✉e-mail: yonghphysics@gmail.com; xxwu@itp.ac.cn; ming.shi@psi.ch

