## [Peer Review File · Nature Communications]

REVIEWER COMMENTS

Reviewer #1 (Remarks to the Author):

The author conducts a comprehensive study of ScV₆Sn₆, a Kagome metal, confirming the presence of a CDW through ARPES and STM, consistent with prior Kagome metal studies. The key novelty lies in the co-existence of multiple phonon modes with the CDW. Temperature-dependent Raman scattering reveals a two-phonon mode at 116 cm⁻¹ above the 92K CDW transition (TCDW), likely stemming from the K-H acoustic branches. Its disappearance below TCDW is attributed to CDW-induced phonon folding. While intriguing, the link between phonon and CDW promotion requires further support. I recommend providing theoretical or experimental evidence or simply revising the claim into "co-existence of phonon and CDW". Besides, the STM measurement shows the disappearance of the CDW pattern well below the transition temperature. The author should explain this inconsistency. Overall, the study merits publication in Nature Communications with some minor revisions.

Reviewer #2 (Remarks to the Author):

This manuscript reports a comprehensive study of the kagome metal ScV₆Sn₆ using STM, ARPES, Raman scattering, and DFT calculations. STM topography visualizes $\sqrt{3}\times\sqrt{3}$ lattice reconstruction. ARPES measurements discover electronic states which are sensitive to the surface termination and identify Dirac cones and VHSs expected for the kagome metal. The authors argue that the VHSs near K points may contribute to nesting and the CDW formation. However, the electronic band structure was observed to be barely affected by the CDW transition. The major findings of the Raman spectroscopy are two modes that the authors argue to be relevant for the CDW. They are assigned as a two-phonon mode and an amplitude mode, which are suggested to be evidence of strong electron-phonon coupling. The authors conclude that the origin of the CDW is mainly driven by the lattice degree of freedom.

Kagome metals offer a playground to explore novel electronic ground states and topological properties. ScV₆Sn₆ has attracted recent research interest as a new system to realize such physics. The current work is timely and provides important details that are essential for understanding this system. I believe this will be a valuable contribution to this field. However, I have the following concerns which I hope the authors can clarify.

1. In the Methods section, it was mentioned that two batches of crystals were studied. Their CDW transition temperatures differ by 12 K. Can the authors comment on the motivation for choosing two batches of crystals? How to explain such a large difference in TCDW? More importantly, have the authors compared the two batches of crystals using STM, ARPES, and Raman?

2. The major part of the work focuses on characterizing the material using ARPES. In terms of clarifying the CDW formation mechanism as the goal of the current work, it is not clear to me what the ARPES data have helped to resolve this issue. The most relevant result appears to be the discovery of K-point-related VHSs, which can be correlated with the lattice reconstruction. But temperature-dependent measurements found negligible change of the band structure due to the CDW transition. Can the authors elaborate on the implications of these results? The absence of change in the electronic band structure is quite surprising, as the resistivity shows a more dramatic change at the CDW transition than that seen in Cs₃VSb₅. How to understand this discrepancy in the temperature dependence of the electronic band structure and electronic transport?

3. The authors assigned the broad feature at about 120 cm⁻¹ as a two-phonon mode. How to rule out other sources for this mode, such as defect-induced scattering? For Cs₃VSb₅, the observed Raman features from different groups clearly differ, which appears to be related with different sample quality. Have the authors compared the Raman response using the two batches of crystals?

4. In the text it was described that this two-phonon mode is suppressed below TCDW. This is not clear in Fig. 4a. Judging from the spectra shown in Fig. S6b, this broad peak sits on a strong background that rises sharply at low energy. Can the authors do fitting analysis on this two-phonon mode to show its suppression below TCDW? What is the origin of the rising background?

5. The assignment of the amplitude mode seems questionable. This mode is extremely weak in the linear scale plot shown in Fig. S6b. It is enhanced by using a log plot in Fig. 4b. I do not think any reliable analysis can be made on such a weak feature. How were the results in Fig. 4e obtained?

6. In Fig. S6b, data for the crossed polarization at 4 K should be used. Including another set of data for parallel and crossed polarizations at 300 K will also be helpful.

Minor issues

Lines 180 and 181: It is difficult to tell what "highlighted by the dashed white curve in Fig. 4a" and "indicated by the dashed red line in Fig. 4a" refer to on the figure. The authors may consider using a different color.

Fig 4a,b: The Raman shift axes are misaligned, as there is a clear shift for the two major modes in the two panels.

Reviewer #3 (Remarks to the Author):

This manuscript describes a systematic investigation of ScV₆Sn₆, a recently discovered topological bilayer kagome crystal with CDW order, employing ARPES, DFT calculations and Raman measurements. Compared to DFT calculations, their ARPES data revealed some features in the band structure of ScV₆Sn₆, like the Dirac-cones and VHSs inherent to the kagome lattice. Additionally, a new VHS near K point was observed. Their Raman measurements unveiled a strong electron-phonon coupling, indicating the potential lattice degree of freedom origin of CDW in ScV₆Sn₆.

The current ARPES spectra in the manuscript didn't provide the evidence or characteristics of CDW in vanadium based 166 compounds such as band folding and band renormalization, which would give insight into the CDW therein. Therefore, I cannot recommend it for publication in Nature Communication. Here are my main comments and suggestions.

In this manuscript, the authors state that "ScV₆Sn₆ tends to cleave along the c-axis, resulting in three surface terminations, namely the Kagome, ScSn₃₂ and Sn₂ layers." However, it should be noted that the V atoms and Sn atoms within the kagome layer are not situated on the same plane, which leading to different chemical environments on its upper and lower sides. One latest theoretical paper (arXiv: 2308.08771) also shows some differences between the two sides of kagome layer. I am interested in whether there are any differences in ARPES or STM measurements for the two sides of the kagome Layer.

I noticed that the definition of surface terminations in recent papers are confused. In sharp contrast to this manuscript, some previous ARPES papers on RV₆Sn₆ materials (e.g., PRL 127, 266401; Nat. Phys. 19, 1135-1142) present a different definition of terminations. The authors are suggested to resolve the surface termination issue more clearly, e.g., to present detailed comparison between the ARPES data and the calculation with the surface state on the different surface terminations.

The authors present a new van Hove singularity (VHS₃) near the K point and the nesting vectors associated with this VHS are in proximity to (1/3, 1/3). I believe this is one of the main discoveries in

this manuscript. On one hand, these VHS3's are located too far away from the Fermi level; on the other hand, the authors did not present a convincing temperature-dependent data on these VHS bands. While this manuscript showcases the ARPES intensity plots along Γ -M-K-M- Γ direction over a wide binding energy range of -1.2 eV, it does not focus on the VHSs. Consequently, identifying the VHSs in the figure 3k and 3l is challenging and tracing band evolution across the CDW phase transition becomes even more arduous. An intensive temperature-dependent measurements is crucial to eliminate possible discrepancies arising from tiny energy shifts or gap opening.

The ARPES spectra show the different dispersion at the different surface termination and the different polarization. However, there is no new information about the 166 compounds. Previous Ref. [Sci. Adv. 8, eadd2024 (2022)] about GdV6Sn6 already discuss the termination and polarization dependent ARPES measurement and provide the similar result about the VHS1 and 2. The ARPES spectra in the manuscript didn't provide the evidence or characteristics of CDW in vanadium based 166 compounds such as band folding and band renormalization, and the fact that can give some insight for the CDW.

In Fig. 4, the authors show the Raman spectra to discuss the phonon mode for the CDW mechanism and compare with the phonon mode in CsV3Sb5. The authors discuss that the CDW in ScV6Sn6 mainly attribute to the phonon mode of Sc and Sn atoms not V atoms. It provides a different CDW mechanism with the CsV3Sb5 which is mainly contributed by the phonon mode of V atoms and electron nesting. What is the relationship between the ARPES and Raman measurements? The authors mentioned that ScV6Sn6 have stronger electron-phonon coupling comparing the case of CsV3Sb5. It implies that the electronic structure of ScV6Sn6 can more clearly show the evidence of the electron-phonon coupling comparing with CsV3Sb5, e.g., kink feature in CsV3Sb5 Ref. [Nat. Commnu. 14, 1945 (2023)]. While, the manuscript didn't show any difference depending on the temperature and the evidence for the strong electron-phonon interaction. Authors need to suggest the relationship between ARPES and Raman results in the manuscript.

A couple of minor comments:

- In Figs. 2 and 3, author use the different color scale for the APRES spectra for raw and second derivative spectra. Authors use the black and white color scale for the second derivative of the ARPES spectra and for the raw data is for the blue and white. But author use the black and white color scale for the raw data in Figs. 3 g, h, l, k and l. There is possibility to give a confuse to general reader. Author should use the same color scale for raw and second derivative data.

- The ARPES measurement taken from the linear vertical polarization can show more clear dispersion for the m-type VHS as shown in Fig. 2k and 3h.

- In Ref. [arXiv:2302.12227], this paper argues that the STM data of Sn surface didn't show the CDW peaks for $\sqrt{3}\times\sqrt{3}$, but the V termination show the CDW peaks and the CDW gap. How about the STM data taken from the V termination of the manuscript?

- It seems that Fig. S4f is not mentioned in either main text or supplementary materials.

Phonon promoted charge density wave in topological kagome metal ScV_6Sn_6

Yong Hu^{1,#,*}, Junzhang Ma^{2,3,4,#}, Yinxiang Li^{5,#}, Yuxiao Jiang⁶, Dariusz Jakub Gawryluk⁷, Tianchen Hu⁸, Jérémie Teyssier⁹, Volodymyr Multian⁹, Zhouyi Yin¹⁰, Shuxiang Xu⁸, Soohyeon Shin⁷, Igor Plokhikh⁷, Xinloong Han¹¹, Nicholas C. Plumb¹, Yang Liu¹², Jia-Xin Yin¹³, Zurab Guguchia¹⁴, Yue Zhao¹⁰, Andreas P. Schnyder¹⁵, Xianxin Wu^{16,*}, Ekaterina Pomjakushina⁷, M. Zahid Hasan⁶, Nanlin Wang^{8,17,18}, and Ming Shi^{1,12,*}

¹*Photon Science Division, Paul Scherrer Institut, CH-5232 Villigen PSI, Switzerland*

²*Department of Physics, City University of Hong Kong, Kowloon, Hong Kong, China*

³*City University of Hong Kong Shenzhen Research Institute, Shenzhen, China*

⁴*Hong Kong Institute for Advanced Study, City University of Hong Kong, Kowloon, Hong Kong, China*

⁵*College of Science, University of Shanghai for Science and Technology, Shanghai, 200093, China*

⁶*Laboratory for Topological Quantum Matter and Advanced Spectroscopy (B7), Department of Physics, Princeton University, Princeton, NJ, USA*

⁷*Laboratory for Multiscale Materials Experiments, Paul Scherrer Institut, CH-5232 Villigen PSI, Switzerland*

⁸*International Center for Quantum Materials, School of Physics, Peking University, Beijing 100871, China*

⁹*Department of Quantum Matter Physics, University of Geneva, 24 Quai Ernest-Ansermet, 1211 Geneva 4, Switzerland*

¹⁰*Institute for Quantum Science and Engineering and Department of Physics, Southern University of Science and Technology of China, Shenzhen, Guangdong 518055, China*

¹¹*Kavli Institute for Theoretical Sciences, University of Chinese Academy of Sciences, Beijing 100190, China*

¹²*Center for Correlated Matter and Department of Physics, Zhejiang University, Hangzhou 310058, China*

¹³*Department of physics, Southern University of Science and Technology, Shenzhen, Guangdong 518055, China*

¹⁴*Laboratory for Muon Spin Spectroscopy, Paul Scherrer Institute, CH-5232 Villigen PSI, Switzerland*

¹⁵*Max-Planck-Institut für Festkörperforschung, Heisenbergstrasse 1, D-70569 Stuttgart, Germany*

¹⁶*CAS Key Laboratory of Theoretical Physics, Institute of Theoretical Physics, Chinese Academy of Sciences, Beijing 100190, China*

¹⁷*Beijing Academy of Quantum Information Sciences, Beijing 100913, China*

¹⁸*Collaborative Innovation Center of Quantum Matter, Beijing 100871, China*

#These authors contributed equally to this work.

**To whom correspondence should be addressed:*

Y.H. (yonghphysics@gmail.com); X.W. (xxwu@itp.ac.cn); M.S. (ming.shi@psi.ch)

Reply to Reviewers' Reports

We express our sincere gratitude to all the Reviewers for their careful review of our manuscript and for recognizing the significance of our work. Their constructive comments and suggestions have been invaluable for improving our manuscript. In this Response, we have addressed all the concerns raised by the Reviewers in a point-by-point manner, and have indicated the corresponding changes made in the revised manuscript. These changes are listed on the last page of this reply letter.

Point-to-Point Response to Reviews

For clarity, the reviewers' original comments are shown in blue italic font.

The authors' responses are shown in black regular font.

Reviewer #1 (Remarks to the Author):

The author conducts a comprehensive study of ScV₆Sn₆, a Kagome metal, confirming the presence of a CDW through ARPES and STM, consistent with prior Kagome metal studies. The key novelty lies in the co-existence of multiple phonon modes with the CDW. Temperature-dependent Raman scattering reveals a two-phonon mode at 116 cm⁻¹ above the 92K CDW transition (T_{CDW}), likely stemming from the K-H acoustic branches. Its disappearance below T_{CDW} is attributed to CDW-induced phonon folding. While intriguing, the link between phonon and CDW promotion requires further support. I recommend providing theoretical or experimental evidence or simply revising the claim into "co-existence of phonon and CDW".

We sincerely thank the Reviewer for his/her meticulous evaluation of our work. The provided comments accurately capture the essential points and significance of our study. We are thankful for the constructive suggestions, which have significantly contributed to the improvement of our manuscript.

Our Raman measurements have revealed a two-phonon mode in the normal state and Raman-active amplitude modes in the CDW phase, indicating strong electron-phonon coupling. The presence of the two-phonon mode in the normal state and the emergence of the amplitude mode in the CDW phase suggest an intimate link between phonon dynamics and CDW promotion.

Fig. R1. Temperature dependence of the A_{1g} mode in ScV_6Sn_6 . **a** Temperature-dependent Raman spectra illustrating the A_{1g} mode. The black arrows indicate the Raman peaks. **b,c** Evolution of frequency (**b**) and linewidth (**c**) for the A_{1g} mode. Notably, around the CDW phase transition temperature, the sum of the linewidth of the two single peaks below T_{CDW} is well above the single peak linewidth above T_{CDW} , suggesting a splitting of the A_{1g} mode in the CDW state.

Furthermore, our latest Raman measurements, featuring improved data quality, have unveiled a novel splitting of the A_{1g} mode in the CDW state (Fig. R1). This observation provides supporting evidence for the modulation of the lattice along the c -axis induced by the CDW, thereby reinforcing the crucial role of electron-lattice coupling in promoting the 3D CDW.

Moreover, subsequent to the submission of our paper [[arXiv:2304.06431](https://arxiv.org/abs/2304.06431)], we have discovered alignment between our conclusion of a phonon-promoted CDW in ScV_6Sn_6 and later inelastic x-ray

(IXS) measurements. These IXS measurements unveil the softening of a flat phonon mode in the kagome structure of ScV_6Sn_6 as it approaches the CDW transition temperature [*arXiv:2304.08197; arXiv:2304.09173*].

In the revised Supplementary Materials, we have included the observed splitting of the A_{1g} mode and extended the corresponding discussions on this finding to provide further support for the link between phonon and CDW promotion.

Besides, the STM measurement shows the disappearance of the CDW pattern well below the transition temperature. The author should explain this inconsistency. Overall, the study merits publication in Nature Communications with some minor revisions.

In our study, we examined two batches of ScV_6Sn_6 samples obtained from distinct research groups, with charge density wave transition temperatures of $T_{\text{CDW}} = 92 \text{ K}$ [33] and $T_{\text{CDW}} = 80 \text{ K}$ [30], respectively. To facilitate a more accessible exploration of the normal state of ScV_6Sn_6 , we conducted STM measurements on the sample with $T_{\text{CDW}} = 80 \text{ K}$. As a result, there is no inconsistency. Additional details regarding these STM measurements are provided in the Method section of the revised manuscript for clarity on this aspect.

Reviewer #2 (Remarks to the Author):

This manuscript reports a comprehensive study of the kagome metal ScV_6Sn_6 using STM, ARPES, Raman scattering, and DFT calculations. STM topography visualizes $\sqrt{3}\times\sqrt{3}$ R30deg lattice reconstruction. ARPES measurements discovers electronic states which are sensitive to the surface termination and identifies Dirac cones and VHSs expected for the kagome metal. The authors argue that the VHSs near K points may contribute to nesting and the CDW formation. However, the electronic band structure was observed to be barely affected the CDW transition. The major findings of the Raman spectroscopy are two modes that the authors argue to be relevant for the CDW. They are assigned as a two-phonon mode and an amplitude mode, which are suggested to be evidence of strong electron phonon coupling. The authors conclude that the origin of the CDW is mainly driven by the lattice degree of freedom.

Kagome metals offer a playground to explore novel electronic ground states and topological properties. ScV_6Sn_6 has attracted recent research interest as a new system to realize such physics. The current work is timely and provides important details that are essential for understanding this system. I believe this will be a valuable contribution to this field. However, I have the following concerns which I hope the authors can clarify.

We would like to express our sincere gratitude to the Reviewer for recognizing the significance of our work. We greatly appreciate his/her constructive comments and suggestions, which have been valuable in helping us to improve our manuscript.

1. In the Methods section, it was mentioned that two batches of crystals were study. Their CDW transition temperatures differ by 12 K. Can the authors comment on the motivation for choosing two batches of crystals? How to explain such a large difference in T_{CDW} ? More importantly, have the authors compared the two batches of crystals using STM, ARPES, and Raman?

The two batches of ScV_6Sn_6 samples were obtained from different research groups. The slightly different crystal growth processes employed by the two groups are likely the origins of the different CDW transitions temperatures.

In our initial study, we measured both batches of crystals using ARPES, STM, and μ SR [arXiv: 2304.06436 (2023)]. As stated in the Method section, 'the quality of ARPES data from the two different TCDW samples is comparable, and the corresponding results appear to be the same; ARPES spectra show negligible differences between the CDW phase and the normal state.' Additionally, our μ SR measurements indicate time-reversal symmetry breaking in both types of crystals [arXiv: 2304.06436 (2023)].

Fig. R2. Temperature dependence of the Raman response in ScV₆Sn₆. **a** Recorded on the second batch ($T_{\text{CDW}} = 92$ K) of ScV₆Sn₆. **b** Comparison of the Raman spectrum at 15 K obtained from two batches of samples.

Recently, we conducted additional Raman measurements on the $T_{\text{CDW}} = 92$ K samples. As illustrated in Fig. R2a, the temperature evolution of the Raman spectra is generally consistent with our previous measurements on the $T_{\text{CDW}} = 80$ K samples (Fig. R2b). Moreover, the new datasets exhibit higher quality, likely owing to improved Raman experimental conditions.

Consequently, there is no significant difference in the physical properties of the two batches of samples. However, in the revised manuscript, we have updated the previous Raman data with the latest Raman results from the $T_{\text{CDW}} = 92$ K samples. Hence, now all the experimental data presented in our work were obtained from the $T_{\text{CDW}} = 92$ K samples, except for the STM topograph measured at 80 K (Fig. 1d), as the $T_{\text{CDW}} = 80$ K samples provide a better opportunity to access the normal states through STM measurements conducted with liquid nitrogen cooling.

2. The major part of the work focuses on characterizing the material using ARPES. In terms of clarifying the CDW formation mechanism as the goal of the current work, it is not clear to me what the ARPES data have helped to resolve this issue. The most relevant result appear to be the discovery of K-point-related VHSs, which can be correlated with the lattice reconstruction. But temperature dependent measurements found negligible change of the band structure due to the CDW transition. Can the authors elaborate on the implications of these results? The absence of change in the electronic band structure is quite surprising, as the resistivity shows a more dramatic change at the CDW transition than that seen in Cs3VSb5. How to understand this discrepancy in the temperature dependence of the electronic band structure and electronic transport?

The primary objective of our paper is to decipher the origin of the CDW order of kagome metals ScV₆Sn₆. To accomplish this, we conducted ARPES measurements to scrutinize the electronic contribution and Raman scattering measurements to assess the contribution from the lattice degree of freedom. From the electronic perspective, as nicely summarized by the Reviewer, our high-resolution ARPES measurements unveiled distinct electronic states, including topologically nontrivial Dirac surface states (TDSSs), Dirac cones and multiple van Hove singularities (VHSs) in the vicinity of

the Fermi level (E_F) (refer to Fig. 3 in the main text). Furthermore, our temperature-dependent ARPES measurements revealed negligible differences between the CDW phase and the normal state. This can be attributed to the noticeable dispersion along the c -direction and the three-dimensional nature of the $\sqrt{3} \times \sqrt{3} \times 3$ CDW order. These experimental outcomes provide crucial insights for understanding the unconventional CDW order in ScV_6Sn_6 .

Firstly, the VHSs near the \bar{K} point could introduce nesting wave vectors close to $(1/3, 1/3)$, matching the observed $\sqrt{3} \times \sqrt{3}$ $R30^\circ$ CDW wave vector. This suggests that electronic correlation might play a role in promoting the in-plane component of CDW order.

Secondly, our polarization-dependent measurements indicate a strong sensitivity of the band structures in ScV_6Sn_6 to the photon polarization (see Fig. 2), reflecting the dominant contributions of $V-d$ orbitals and their multi-orbital nature near E_F .

Thirdly, our photon-energy dependent measurements reveal a relatively obvious three-dimensionality in the electronic structure of ScV_6Sn_6 (refer to Fig. 2 and Fig. S2), a key difference compared to CsV_3Sb_5 .

Fourthly, the negligible differences between the V-kagome bands in the CDW phase and normal state suggest weak potential band reconstruction and folding effect in ScV_6Sn_6 , implying a small distortion of V atoms inside the CDW phase.

The above findings stand in sharp contrast to observations in the 2×2 CDW phase of CsV_3Sb_5 , where the electronic structure is relatively two-dimensional and the VHSs are positioned around the Brillouin zone boundary (M point). Additionally, CsV_3Sb_5 exhibits pronounced band reconstructions near E_F in its CDW state, as the CDW pattern mainly involves the distortion of the kagome V atoms.

It is essential to highlight that the marginal change in the band structure is consistent with findings from single crystal x-ray diffraction studies [*PRX* 11, 041030 (2021); *PRL* 129, 216402 (2022)]. In CsV_3Sb_5 , vanadium atoms displace in the plane by 0.009–0.085 Å, forming either the Star of David or tri-hexagonal arrangement. In contrast, in ScV_6Sn_6 , Sc and Sn1 exhibit the largest modulated displacements, displacing up to 0.16 Å along the c -axis and forming Sn1 dimers. While vanadium atoms appear to have a much weaker response, displacing only 0.004–0.006 Å. Therefore, given that the CDW order in ScV_6Sn_6 primarily involves the displacement of Sc and Sn, with V atoms showing negligible distortion, the absence of a significant change in the electronic structure near E_F is anticipated. The crucial role of the lattice degrees of freedom in promoting the CDW is also further supported by our Raman scattering measurements.

While the V-kagome bands near E_F show negligible differences between the CDW phase and the normal state, electronic states contributed by Sc and Sn atoms with noticeable CDW-induced distortion may also play a role in electronic transport. The first order CDW transition is expected to result in a dramatic change in resistance.

3. The authors assigned the broad feature at about 120 cm^{-1} as a two-phonon mode. How to rule out other source for this mode, such as defect induced scattering? For Cs_3VSb_5 , the observed Raman features from different groups clearly differ, which appears to be related with different sample quality. Have the authors compared the Raman response using the two batches of crystals?

We conducted extensive temperature-dependent Raman measurements on various batches of single crystals. As illustrated in Figs. R2a and R3a, these updated data confirm the existence of the two-phonon mode and its consistency across different crystal batches. The persistence of the two-phonon

mode above the CDW transition temperature and its disappearance below T_{CDW} suggest a close link between phonon dynamics and CDW promotion, ruling out the defect origin and emphasizing the intrinsic nature of the observed phonon mode. Additionally, our theoretical calculations [Fig. 4f(ii)] reveal the presence of flat acoustic longitudinal modes in the K - H path, providing a plausible explanation for the suggested two-phonon mode.

Fig. R3. Temperature dependence of the two-phonon mode in ScV₆Sn₆. **a** Raman spectra measured at 15 K (blue) and 300 K (red) along with their fittings (black), obtained from the second batch ($T_{CDW} = 92$ K) of ScV₆Sn₆. **b** Fitted Raman intensities of the two-phonon mode.

4. In the text it was described that this two-phono mode is suppressed below TCDW. This is not clear in Fig. 4a. Judging from the spectra shown in Fig. S6b, this broad peak sits on a strong background that rises sharply at low energy. Can the authors do fitting analysis on this two-phonon mode to show its suppression below TCDW? What is the origin of the rising background?

We conducted detailed temperature-dependent Raman measurements on various batches of single crystals. The improved data quality in Figs. R1 and R2 highlights a more pronounced two-phonon mode with only an electronic background, which remains relatively constant. In Fig. R3b, the fitted intensity of the two-phonon mode exhibits a clear suppression as the temperature approaches T_{CDW} .

5. The assignment of the amplitude mode seems questionable. This mode is extremely weak in the linear scale plot shown in Fig. S6b. It is enhanced by using a log plot in Fig. 4b. I do not think any reliable analysis can be made on such a weak feature. How were the results in Fig. 4e obtained?

In the revised manuscript, we have replaced the previous Raman data with the new dataset. The amplitude mode is now clearly revealed and can be reliably analyzed using the reffit software [50] (<https://reffit.ch/>).

6. In Fig. S6b, data for the crossed polarization at 4 K should be used. Including another set of data for parallel and crossed polarizations at 300 K will also be helpful.

In the revised Supplementary Fig. S6 (also Fig. R4), we have presented a comparison of Raman spectra obtained with parallel and crossed polarizations at 15 K and 300 K, respectively.

Minor issues

Lines 180 and 181: It is difficult to tell what "highlighted by the dashed white curve in Fig. 4a" and "indicated by the dashed red line in Fig. 4a" refer to on the figure. The authors may consider using a different color.

In the revised manuscript Fig.4a has been modified.

Fig 4a,b: The Raman shift axes are misaligned, as there is a clear shift for the two major modes in the two panels.

In response to the identified issue, we have rectified the misalignment in the revised manuscript.

Fig. R4. Polarization dependence of the Raman response in ScV₆Sn₆. Raman spectra measured on the *ab*-plane in parallel (blue) and crossed (red) polarizations at 15 K (lower panel) and 300 K (upper panel).

Reviewer #3 (Remarks to the Author):

This manuscript describes a systematic investigation of ScV₆Sn₆, a recently discovered topological bilayer kagome crystal with CDW order, employing ARPES, DFT calculations and Raman measurements. Compared to DFT calculations, their ARPES data revealed some features in the band structure of ScV₆Sn₆, like the Dirac-cones and VHSs inherent to the kagome lattice. Additionally, a new VHS near K point was observed. Their Raman measurements unveiled a strong electron-phonon coupling, indicating the potential lattice degree of freedom origin of CDW in ScV₆Sn₆. The current ARPES spectra in the manuscript didn't provide the evidence or characteristics of CDW in vanadium based 166 compounds such as band folding and band renormalization, which would give insight into the CDW therein. Therefore, I cannot recommend it for publication in Nature Communication. Here are my main comments and suggestions.

We sincerely thank the Reviewer for the careful reviewing our paper. His/her comments capture the essential points of our work. We also appreciate his/her constructive feedback and suggestions provided, which have been instrumental in improving our manuscript.

We would like emphasize that our primary objective is to decipher the origin of the CDW order of the kagome metals ScV₆Sn₆. To achieve this, we conducted ARPES measurements to scrutinize the electronic contribution and Raman scattering measurements to assess the contribution from the lattice degree of freedom. Our detailed analysis of ARPES and Raman data suggests that electron phonon coupling may be the primary driving force for the CDW.

Our high-resolution ARPES measurements unveiled distinct electronic states, including topologically nontrivial Dirac surface states (TDSSs), Dirac cones and multiple van Hove singularities (VHSs) in the

vicinity of the Fermi level (E_F) (refer to Fig. 3 in the main text). Furthermore, temperature-dependent ARPES measurements revealed negligible differences between the CDW phase and the normal state. This can be attributed to the noticeable dispersion along the c -direction and the three-dimensional nature of the $\sqrt{3} \times \sqrt{3} \times 3$ CDW order. These experimental outcomes provide crucial insights for understanding the unconventional CDW order in ScV_6Sn_6 . Here, we summarize our main findings and the implications of our ARPES measurements.

Firstly, the VHSs near the \bar{K} point could introduce nesting wave vectors close to $(1/3, 1/3)$, matching the observed $\sqrt{3} \times \sqrt{3}$ $R30^\circ$ CDW wave vector. This suggests that electronic correlation might play a role in promoting the in-plane component of the CDW order.

Secondly, our polarization-dependent measurements indicate a strong sensitivity of the band structures in ScV_6Sn_6 to the photon polarization (see Fig. 2), reflecting the dominant contributions of $V-d$ orbitals and their multi-orbital nature near E_F .

Thirdly, our photon-energy dependent measurements reveal a relatively obvious three-dimensionality in the electronic structure of ScV_6Sn_6 (refer to Fig. 2 and Fig. S2), a key difference compared to CsV_3Sb_5 .

Fourthly, the negligible differences between the V -kagome bands in the CDW phase and normal state suggest weak potential band reconstruction and folding effect in ScV_6Sn_6 , implying a small distortion of V atoms inside the CDW phase.

The above findings stand in sharp contrast to observations in the 2×2 CDW phase of CsV_3Sb_5 , where the electronic structure is relatively two-dimensional and the VHSs are positioned around the Brillouin zone boundary (M point). Additionally, CsV_3Sb_5 exhibits pronounced band reconstructions near E_F in its CDW state, as the CDW pattern mainly involves the distortion of the kagome V atoms. Our ARPES measurements indicate that Fermi surface nesting in ScV_6Sn_6 is less pronounced than CsV_3Sb_5 .

Our Raman measurements, on the other hand, revealed the presence of a two-phonon mode in the normal state and Raman-active amplitude modes in the CDW phase, indicating a strong electron-phonon coupling. Therefore, the combination of the ARPES and Raman measurements emphasizes the dominant role of lattice degrees of freedom in promoting the CDW in ScV_6Sn_6 . Our conclusion of a phonon promoted CDW in ScV_6Sn_6 aligns with later inelastic x-ray (IXS) measurements that reveal the softening of a flat phonon mode in the kagome ScV_6Sn_6 approaching the CDW transition temperature [arXiv:2304.08197; arXiv:2304.09173].

Below are our responses to the issues raised by the Reviewer together with the changes that we made in the revised paper to address the Reviewer's concerns.

In this manuscript, the authors state that “ ScV_6Sn_6 tends to cleave along the c -axis, resulting in three surface terminations, namely the Kagome, ScSn_{32} and Sn_2 layers.” However, it should be noted that the V atoms and Sn atoms within the kagome layer are not situated on the same plane, which leading to different chemical environments on its upper and lower sides. One latest theoretical paper (arXiv:2308.08771) also shows some differences between the two sides of kagome layer. I am interested in whether there are any differences in ARPES or STM measurements for the two sides of the kagome Layer.

We appreciate the constructive comments from the Reviewer. Structurally, as illustrated in Fig. R5a, there are four potential surface terminations in RV_6Sn_6 (where R represents a rare-earth element),

namely V_3Sn (kagome), $ScSn^{3_2}$, Sn^2 , and SnV_3 terminations (Fig. R5). However, experimentally distinguishing between them is highly challenging.

In ARPES measurements, such as in the kagome metal $FeSn$, it has been noted that Sn atoms in different local environments contribute $4d$ peaks with distinct binding energies [*Nat. Mater.* 19, 163–169 (2020)]. Employing this strategy, XPS spectra of the $Sn-4d$ peaks are utilized to differentiate the surface terminations of ScV_6Sn_6 . Notably, in ScV_6Sn_6 , the Sn atoms have a more complex local environment than those in $FeSn$. Using a small beam spot, three sets of electronic structure in ScV_6Sn_6 were revealed. Given the similarity in local environments of Sn atoms in the Sn^2 and SnV_3 terminations, especially when extra Sn forms on the Sn^2 and SnV_3 terminations, causing the Sn^2 and SnV_3 terminations to appear experimentally mixed and indistinguishable, we assign the three sets of electronic structure to the vanadium kagome, $ScSn^{3_2}$, and Sn terminations (Fig. R5b).

In comparison to ARPES, STM can more directly differentiate surface terminations when atomic resolution is achieved. The SnV_3 termination exhibits a triangular lattice, while the kagome termination (V_3Sn) has a hexagonal lattice. Our STM measurements have revealed both a triangular lattice with atomic resolution and a hexagonal lattice (Fig. R5c).

Fig. R5. Crystal structure and topographic characterization of RV_6Sn_6 illustrating potential surface terminations. **a** The bulk crystal structure of TbV_6Sn_6 , adapted from the theoretical paper (*arXiv*: 2308.08771). **b** Three plausible surface terminations of ScV_6Sn_6 suggested in ARPES measurements. **c** STM topographs of ScV_6Sn_6 showing a honeycomb (i), a triangular (ii), and a hexagonal (iii) lattice. **The clear atomic resolution of the honeycomb and triangular lattice leads us to assign them to the Sn^2 and SnV_3 termination, respectively. The hexagonal lattice without atomic resolution can be either Kagome termination or $ScSn$ termination. Considering that for $ScSn$ termination, there is a Sc atom in the center of the hexagon which we do not observe here, we suggest that the hexagonal lattice (iii) is more likely to be the Kagome termination. The topographic image on the Kagome termination of ScV_6Sn_6 resembles the Kagome terminations identified in**

the existing papers [*Nat. Phys.* 18, 644–649 (2022); *PRL* 129, 166401 (2022); *Nat. Commun.* 12, 4269 (2021); *Nat. Commun.* 11, 5613 (2020)], where the absence of atomic resolution can be due to the close atomic distance.

I noticed that the definition of surface terminations in recent papers are confused. In sharp contrast to this manuscript, some previous ARPES papers on RV_6Sn_6 materials (e.g., *PRL* 127, 266401; *Nat. Phys.* 19, 1135-1142) present a different definition of terminations. The authors are suggested to resolve the surface termination issue more clearly, e.g., to present detailed comparison between the ARPES data and the calculation with the surface state on the different surface termination ns .

Fig. R6. Topologically nontrivial Dirac surface states in RV_6Sn_6 . **a, b** DFT electronic structure on GdSn layer (a) and V kagome layer (b) of GdV_6Sn_6 , with bulk states and surface states (SS) decomposed. **c** Experimental TDSs on pristine (i) and K-dosed (ii) surfaces of GdV_6Sn_6 . Data in (a-c) are adapted from our previous paper [*Sci. Adv.* 8, eadd2024 (2022)]. **d** DFT band dispersions for the ScSn layer (i), V_3Sn layer (ii), Sn layer (iii), and SnV_3 layer (iv) of ScV_6Sn_6 .

In GdV_6Sn_6 , the authors of the *PRL* paper [*PRL* 127, 266401 (2022)] assigned surface terminations according to the Sn 4d core level by analogy with the case in FeSn. They argue that the different local environments for Sn atoms with or without adjacent Gd atoms contributes the multiple Sn surface peaks in the XPS measurements, as described in the Figure caption of Figure 2 of the paper. However, unlike the kagome metals FeSn, in GdV_6Sn_6 the Sn atoms close to the kagome termination have a more complex local environment than the Sn atoms in the $GdSn_2$ termination. Therefore, in principle, the XPS spectra showing more Sn-4d peaks should be associated with the kagome termination. Moreover, as discussed in Section S3 of the Supplementary Materials for our previous work [*Sci. Adv.* 8, eadd2024 (2022)], the experimentally observed TDSs (Fig. R6c) are in excellent agreement with the DFT calculations for the $GdSn_2$ layer (Figs. R6a and 6b), strongly supporting our assignment of termination.

In ScV_6Sn_6 , the experimental (Fig. 2j, Figs. 3a and 3b) and theoretical (Fig. R6d) TDSs bear a striking resemblance to the TDSs in GdV_6Sn_6 compound (Figs. R6a-c). As previously established in GdV_6Sn_6 [*Sci. Adv.* 8, eadd2024 (2022)], we assign the surface termination exhibiting the TDSs to the ScSn layer. Furthermore, based on the Fermi surface topology (Fig. S1 in the

Supplementary Materials), we suggest the other two terminations belong to the kagome layer and Sn layer, respectively.

The authors present a new van Hove singularity (VHS3) near the K point and the nesting vectors associated with this VHS are in proximity to (1/3, 1/3). I believe this is one of the main discoveries in this manuscript. On one hand, these VHS3's are located too far away from the Fermi level; on the other hand, the authors did not present a convincing temperature-dependent data on these VHS bands. While this manuscript showcases the ARPES intensity plots along Γ -M-K-M- Γ direction over a wide binding energy range of -1.2 eV, it does not focus on the VHSs. Consequently, identifying the VHSs in the figure 3k and 3l is challenging and tracing band evolution across the CDW phase transition becomes even more arduous. An intensive temperature-dependent measurements is crucial to eliminate possible discrepancies arising from tiny energy shifts or gap opening.

As highlighted earlier, the primary message we aim to convey from our study is the pivotal role of lattice degrees of freedom in promoting the CDW in ScV₆Sn₆. Nonetheless, the discovery of VHS3 near the K point is intriguing, as it could introduce nesting wave vectors close to (1/3,1/3), aligning with the observed $\sqrt{3} \times \sqrt{3}$ R30° CDW wave vector. We acknowledge the Reviewer's point that VHS3 near the K point is not in close proximity to the Fermi level. On the other hand, considering that the CDW order primarily involves the displacement of Sc and Sn atoms, with V atoms showing negligible distortion, the band reconstruction and folding on V-*d* orbitals, including the VHS3 bands, are expected to be very weak. In this context, VHS3 may not directly contribute to the CDW. However, it could potentially play a role in promoting the in-plane component of CDW order, particularly given the noticeable coupling between V and Sc/Sn atoms.

Fig. R7. Electronic structure comparison in ScV₆Sn₆. a Temperature-dependent measurements of the band structure along the $\bar{\Gamma} - \bar{K}$ direction, measured below T_{CDW} at 20 K (a) and above T_{CDW} at 130 K (b) with 74 eV circularly polarized light. The red and orange dashed lines indicate the VHS band bottoms.

Given that the CDW transition in ScV₆Sn₆ is characterized as a first-order phase transition, as validated by x-ray diffraction (XRD) [PRL 129, 216402 (2022)] and optical measurements [PRB 107, 165119 (2023); PRB 108, 205118 (2023)], a direct comparison of ARPES spectra below and above T_{CDW} can provide insights into the potential impact of the CDW on the electronic structure. As depicted in Fig. R7, a closer examination of the band structure collected below and above T_{CDW} reveals negligible differences. Moreover, our STM measurements also do not show a clear gap opening in the CDW phase (Fig. R8c). The absence of an evident energy shift or gap opening in the V-*d* orbitals can be ascribed to the minimal distortion present in the CDW phase.

The ARPES spectra show the different dispersion at the different surface termination and the different polarization. However, there is no new information about the 166 compounds. Previous Ref. [Sci. Adv. 8, eadd2024 (2022)] about GdV₆Sn₆ already discuss the termination and polarization dependent ARPES measurement and provide the similar result about the VHS1 and 2. The ARPES spectra in the manuscript didn't provide the evidence or characteristics of CDW in vanadium based 166 compounds such as band folding and band renormalization, and the fact that can give some insight for the CDW. The negligible differences in the V-kagome bands between the CDW phase and normal state, as indicated by our ARPES measurements, suggest a weak band reconstruction and folding effect near E_F , indicating a small distortion of V atoms in ScV₆Sn₆ inside the CDW phase. These findings imply that Fermi surface nesting may not be the primary driving force for the CDW in ScV₆Sn₆.

It is essential to highlight that the marginal change in the band structure is consistent with findings from single crystal XRD studies [PRX 11, 041030 (2021); PRL 129, 216402 (2022)]. In CsV₃Sb₅, vanadium atoms displace in the plane by 0.009–0.085 Å, forming either the Star of David or tri-hexagonal arrangement. In contrast, in ScV₆Sn₆, Sc and Sn1 exhibit the largest modulated displacements, displacing up to 0.16 Å along the *c*-axis and forming Sn1 dimers. While vanadium atoms appear to have a much weaker response, displacing only 0.004–0.006 Å. Therefore, given that the CDW order in ScV₆Sn₆ primarily involves the displacement of Sc and Sn, with V atoms showing negligible distortion, the absence of a significant change in the electronic structure near E_F is anticipated.

In Fig. 4, the authors show the Raman spectra to discuss the phonon mode for the CDW mechanism and compare with the phonon mode in CsV₃Sb₅. The authors discuss that the CDW in ScV₆Sn₆ mainly attribute to the phonon mode of Sc and Sn atoms not V atoms. It provides a different CDW mechanism with the CsV₃Sb₅ which is mainly contributed by the phonon mode of V atoms and electron nesting. What is the relationship between the ARPES and Raman measurements? The authors mentioned that ScV₆Sn₆ have stronger electron-phonon coupling comparing the case of CsV₃Sb₅. It implies that the electronic structure of ScV₆Sn₆ can more clearly show the evidence of the electron-phonon coupling comparing with CsV₃Sb₅, e.g., kink feature in CsV₃Sb₅ Ref. [Nat. Commnu. 14, 1945 (2023)]. While, the manuscript didn't show any difference depending on the temperature and the evidence for the strong electron-phonon interaction. Authors need to suggest the relationship between ARPES and Raman results in the manuscript.

As emphasized earlier, our systematic ARPES measurements suggest a weak band reconstruction and folding effect in ScV₆Sn₆, indicating a small distortion of V atoms. These findings imply that Fermi surface nesting may not be the dominant driving force behind the CDW ScV₆Sn₆. Our Raman measurements, on the other hand, reveal a two-phonon mode in the normal state and Raman-active amplitude modes in the CDW phase, indicating strong electron-phonon coupling. Thus, the combined insights from the ARPES and Raman measurements underscore the dominant role of lattice degrees of freedom in promoting the CDW in ScV₆Sn₆.

In ScV₆Sn₆, the CDW order primarily involves the displacement of Sc and Sn, while V atoms display negligible distortion. This indicates that the strong electron phonon coupling arises from the Sc and Sn-related phonons, rather than the V phonons. Therefore, observing a kink feature in the electronic bands dominated by V-*d* orbitals near E_F is challenging. This is in stark contrast to the case of CsV₃Sb₅, where vanadium atoms exhibit significant modulated displacements, and spectroscopic evidence for electron-phonon coupling is observable in the kagome bands that originate primarily from the V-*d* orbitals.

In the revised manuscript, we have refined the discussion section of the manuscript to underscore the pivotal insights from our ARPES results.

A couple of minor comments:

- In Figs. 2 and 3, author use the different color scale for the APRES spectra for raw and second derivative spectra. Authors use the black and white color scale for the second derivative of the ARPES spectra and for the raw data is for the blue and white. But author use the black and white color scale for the raw data in Figs. 3 g, h, l, k and l. There is possibility to give a confuse to general reader. Author should use the same color scale for raw and second derivative data.

We appreciate the careful review and suggestions provided by the Reviewer. In the revised manuscript, we have implemented a consistent color scale for both raw and second derivative band dispersions. To distinguish Fermi surface maps from band dispersions, we believe that it would be good to use a different color scale for the Fermi surface maps.

- The ARPES measurement taken from the linear vertical polarization can show more clear dispersion for the m-type VHS as shown in Fig. 2k and 3h.

We agree with the Reviewer that the electron branch of VHS2 along the $\bar{\Gamma} - \bar{M}$ direction is clearly revealed by linear vertical polarization [Fig. 2k(ii)]. In Fig. 3h, the hole branch of VHS2 along the $\bar{\Gamma} - \bar{K}$ direction is presented under circular polarization. Furthermore, the band dispersion along the $\bar{\Gamma} - \bar{K} - \bar{M} - \bar{\Gamma}$ direction, as depicted in Fig. 3d, provides additional support for the VHS nature. We are confident that these results unequivocally demonstrate the existence of VHS2 in our study.

Fig. R8. Termination-dependent STM topographs of ScV_6Sn_6 . **a** STM topograph of Sn^2 termination measured at 1 K (i) and associated Fourier transform (ii). **b** Same data as in (a), but for the kagome termination. **c** Average dI/dV spectra over the Sn^2 (red curve) and kagome (blue curve) termination.

- In Ref. [arXiv:2302.12227], this paper argues that the STM data of Sn surface didn't show the CDW peaks for $\sqrt{3}\times\sqrt{3}$, but the V termination show the CDW peaks and the CDW gap. How about the STM data taken from the V termination of the manuscript?

Figures R8a and 8b depict the Fourier transform images on the Sn and V terminations. On the Sn termination, the $\sqrt{3}\times\sqrt{3}$ CDW peaks are unambiguously resolved, while they are absent on the V termination. The differential conductance on the Sn and V terminations is illustrated in Fig. R8c. There is a suppression of the density of states near the Fermi level on both terminations. However, due to the mild nature of the suppression, we cannot conclusively identify it as a CDW gap.

These observations align with the findings from XDR measurements [*PRL* 129, 216402 (2022)] and DFT calculations [*arXiv*:2302.07922 (2023)], indicating that the CDW order in ScV₆Sn₆ involves primarily the displacement of Sc and Sn, while V atoms exhibit negligible distortion.

- It seems that Fig. S4f is not mentioned in either main text or supplementary materials.

In the revised Supplementary material, we have expanded the discussion of Fig. S4f.

Summary of Modifications

In the revised manuscript, the modifications are highlighted in red font.

1. We have updated previous Raman data with the latest Raman results from the $T_{\text{CDW}} = 92 \text{ K}$ samples.
2. As suggested by Reviewer #1, we have incorporated supplementary Raman results into Fig. S6 of the revised Supplementary Materials, further reinforcing the connection between phonon dynamics and the CDW promotion.
3. In light of the Reviewers' comments, we have refined the discussion section of the manuscript to emphasize the pivotal insights derived from our ARPES results.

REVIEWER COMMENTS

Reviewer #1 (Remarks to the Author):

The author has responded to all my questions and I support the publication ASIS.

Reviewer #2 (Remarks to the Author):

The authors have addressed most of my concerns. The newly added discussions help to clarify the CDW mechanism. With improved quality for the Raman data, the related discussions are now more convincing. However I still have one concern about the assignment of the amplitude mode A3. From the lowest measured temperature to 90 K, the peak shift of this mode is less than 1% of its frequency, and the linewidth is always less than 1% of the mode frequency. For a typical amplitude mode, much larger peak shift (ideally approaching zero following a BCS-type order parameter) and broadening are anticipated. From the revised Fig. 4a, the A3, A4, and A5 modes are not very different in their temperature dependence. I believe the authors should be more careful about making this claim of amplitude mode, unless it is supported by more convincing arguments.

Reviewer #3 (Remarks to the Author):

The authors have thoroughly examined our commits and provided meticulous responses. They have summarized their key findings of their systematic and scientific AREPS measurements in the reply. I am content with the authors' explanation regarding the terminations in this system. However, I am afraid that I still harbor reservations concerning the suitability of publishing this manuscript in Nature Communications.

1、 The majority of the manuscript is dedicated to describe ARPES results on Kagome metal ScV_6Sn_6 , but it remains unclear what new information these measurements can provide. Their findings, including the TDSSs, VHSs and Dirac cones, appear to be transplants from the authors' previous studies on GdV_6Sn_6 [Sci. Adv. 8, eadd2024 (2022)] to ScV_6Sn_6 . These findings do not explain why ScV_6Sn_6 is a unique case within the RV_6Sn_6 family.

2、 The authors indicated that the VHSs near the K point could introduce nesting wave vector that aligning with the observed CDW wave vector, and it suggests that electronic correlation might play a role in promoting the in-plane component of the CDW order. However, they do not directly address how a nesting wave vector from a VHS far away from the Fermi level contributes to the CDW. The authors said it could potentially play a role in promoting the in-plane component of CDW order, particularly given the noticeable coupling between V and Sc/Sn atoms. I think a more detailed theoretical explanation might be necessary if the authors insist in this point in the main text.

3、 The temperature-dependent ARPES measurements show negligible differences between the V-kagome bands in the CDW phase and normal state. While acknowledging the potential significance of negative findings in scientific investigations, I am uncertain whether these results possess sufficient importance to warrant publication in high-impact journals such as Nature Communications since several previous ARPES works have reported the similar result on that. Consequently, I would like to defer this question to the editor. For the information, I noted that some latest ARPES studies on ScV₆Sn₆ revealed energy gaps at the Fermi level [Arxiv 1304.11820; Commun Mater 4, 103 (2023)]. These results appear to diverge from those reported by the authors.

Phonon promoted charge density wave in topological kagome metal ScV_6Sn_6

Yong Hu^{1,#,*}, Junzhang Ma^{2,3,4,#}, Yinxiang Li^{5,#}, Yuxiao Jiang⁶, Dariusz Jakub Gawryluk⁷, Tianchen Hu⁸, Jérémie Teyssier⁹, Volodymyr Multian⁹, Zhouyi Yin¹⁰, Shuxiang Xu⁸, Soohyeon Shin⁷, Igor Plokhikh⁷, Xinlong Han¹¹, Nicholas C. Plumb¹, Yang Liu¹², Jia-Xin Yin¹³, Zurab Guguchia¹⁴, Yue Zhao¹⁰, Andreas P. Schnyder¹⁵, Xianxin Wu^{16,*}, Ekaterina Pomjakushina⁷, M. Zahid Hasan⁶, Nanlin Wang^{8,17,18}, and Ming Shi^{1,12,*}

¹Photon Science Division, Paul Scherrer Institut, CH-5232 Villigen PSI, Switzerland

²Department of Physics, City University of Hong Kong, Kowloon, Hong Kong, China

³City University of Hong Kong Shenzhen Research Institute, Shenzhen, China

⁴Hong Kong Institute for Advanced Study, City University of Hong Kong, Kowloon, Hong Kong, China

⁵College of Science, University of Shanghai for Science and Technology, Shanghai, 200093, China

⁶Laboratory for Topological Quantum Matter and Advanced Spectroscopy (B7), Department of Physics, Princeton University, Princeton, NJ, USA

⁷Laboratory for Multiscale Materials Experiments, Paul Scherrer Institut, CH-5232 Villigen PSI, Switzerland

⁸International Center for Quantum Materials, School of Physics, Peking University, Beijing 100871, China

⁹Department of Quantum Matter Physics, University of Geneva, 24 Quai Ernest-Ansermet, 1211 Geneva 4, Switzerland

¹⁰Institute for Quantum Science and Engineering and Department of Physics, Southern University of Science and Technology of China, Shenzhen, Guangdong 518055, China

¹¹Kavli Institute for Theoretical Sciences, University of Chinese Academy of Sciences, Beijing 100190, China

¹²Center for Correlated Matter and Department of Physics, Zhejiang University, Hangzhou 310058, China

¹³Department of physics, Southern University of Science and Technology, Shenzhen, Guangdong 518055, China

¹⁴Laboratory for Muon Spin Spectroscopy, Paul Scherrer Institute, CH-5232 Villigen PSI, Switzerland

¹⁵Max-Planck-Institut für Festkörperforschung, Heisenbergstrasse 1, D-70569 Stuttgart, Germany

¹⁶CAS Key Laboratory of Theoretical Physics, Institute of Theoretical Physics, Chinese Academy of Sciences, Beijing 100190, China

¹⁷Beijing Academy of Quantum Information Sciences, Beijing 100913, China

¹⁸Collaborative Innovation Center of Quantum Matter, Beijing 100871, China

#These authors contributed equally to this work.

*To whom correspondence should be addressed:

Y.H. (yonghphysics@gmail.com); X.W. (xxwu@itp.ac.cn); M.S. (ming.shi@psi.ch)

Reply to Reviewers' Reports

We extend our sincere appreciation to the Reviewers for their positive feedback and constructive suggestions. Their insightful comments have been invaluable in refining our manuscript. In this response, we have thoroughly addressed all raised concerns, incorporating the necessary changes in the revised manuscript, as detailed on the final page of this reply letter.

Point-to-Point Response to Reviews

For clarity, the reviewers' original comments are shown in blue italic font.

The authors' responses are shown in black regular font.

~~~~~

### Reviewer #1 (Remarks to the Author):

~~~~~

The author has responded to all my questions and I support the publication ASIS.

We thank the Reviewer for his/her positive feedback and support for publishing our work in *Nature Communications*.

~~~~~

### Reviewer #2 (Remarks to the Author):

~~~~~

The authors have addressed most of my concerns. The newly added discussions help to clarify the CDW mechanism. With improved quality for the Raman data, the related discussions are now more convincing. However I still have one concern about the assignment of the amplitude mode A₃. From the lowest measured temperature to 90 K, the peak shift of this mode is less than 1% of its frequency, and the linewidth is always less than 1% of the mode frequency. For a typical amplitude mode, much larger peak shift (ideally approaching zero following a BCS-type order parameter) and broadening are anticipated. From the revised Fig. 4a, the A₃, A₄, and A₅ modes are not very different in their temperature dependence. I believe the authors should be more careful about making this claim of amplitude mode, unless it is supported by more convincing arguments.

We sincerely appreciate the positive comments and constructive suggestions provided by the Reviewer.

In agreement with the Reviewer's observations, we acknowledge that the peak shift of the A₃ mode is relatively small, and the A₃, A₄, and A₅ modes exhibit a similar temperature dependence (Fig. 4d in the main text, also Fig. R1a). Given the relatively stronger intensity of the A₃ mode, our primary focus in the manuscript is on this particular mode. Several potential scenarios could explain the appearance of these new modes at low temperatures:

- 1) **Modification of lattice symmetry:** This could allow for new modes or the splitting of existing ones. However, there is no evidence of other phase transitions reported by other techniques. Moreover, if it were a structural phase transition, additional modes would likely emerge from existing ones, which is not observed here.
- 2) **Zone folding:** This often results from an overlapping symmetry in the crystal lattice, such as charge density waves (CDW) or long-range magnetic order. In our case, it is likely that some of these modes arise from zone folding, given their proximity to optical phonon branches and their narrow width.
- 3) **Modes associated with lattice modulation itself:** These are new modes linked to the CDW. They can be of amplitude origin (equivalent to transverse optical modes) or in phase (equivalent to acoustic phonons). Both types are, in principle, visible due to the reduction of the Brillouin zone.

We have assigned new modes, such as the A₃ mode, as amplitude modes because they exhibit a soft mode behavior, a strong characteristic of amplitude collective excitations. While zone folding might contribute, these new modes in ScV₆Sn₆ behave more like amplitude modes with a very long

coherence length. Depending on crystal quality and CDW stability, these amplitude modes may have very similar coherence to normal phonons, resulting in comparable widths. Additionally, these new modes with relatively weak softening in ScV_6Sn_6 (Fig. R1a) closely resemble the amplitude modes suggested in CsV_3Sb_5 (Fig. R1b), where the peak shift is similarly minimal. The absence of complete softening to zero frequency could be related to the first-order CDW transition in these kagome metals.

Fig. R1. Temperature dependence of the Raman response in ScV_6Sn_6 and CsV_3Sb_5 . a,b Temperature-dependent Raman spectra for the A_3 mode in ScV_6Sn_6 (a) and the amplitude mode in CsV_3Sb_5 (b) The data in (b) are adapted from Ref. 42 [*Nat. Commun.* **13**, 3461 (2022)].

In addressing the assignment of the A_3 mode in the revised manuscript, we have moderated our language to allow for alternative interpretations. Achieving unambiguous identification would require further modeling, a scope beyond the present paper. Importantly, concerning the conclusions presented in our paper, distinguishing between an amplitude mode and zone folding is not consequential, as both scenarios showcase robust electron-lattice coupling.

In the revised Supplementary Materials, we have added extra sentences in Section 6 to explore potential scenarios that could elucidate the emergence of new modes at low temperatures.

~~~~~  
**Reviewer #3 (Remarks to the Author):**  
 ~~~~~

*The authors have thoroughly examined our commits and provided meticulous responses. They have summarized their key findings of **their systematic and scientific AREPS measurements** in the reply. I am content with the authors' explanation regarding the terminations in this system.*

We express our gratitude to the Reviewer for revisiting our paper and providing the positive comments. Below, please find our responses to the concerns raised by the Reviewer.

However, I am afraid that I still harbor reservations concerning the suitability of publishing this manuscript in Nature Communications.

*1、 The majority of the manuscript is dedicated to describe ARPES results on Kagome metal ScV_6Sn_6 , but it remains unclear what new information these measurements can provide. Their findings, including the TDSSs, VHSs and Dirac cones, appear to be transplants from the authors' previous studies on GdV_6Sn_6 [*Sci. Adv.* **8**, eadd2024 (2022)] to ScV_6Sn_6 . These findings do not explain why ScV_6Sn_6 is a unique case within the RV_6Sn_6 family.*

As highlighted in our previous Response letter, the primary objective of our present work is to decipher the origin of the charge density wave (CDW) order of the kagome metals ScV_6Sn_6 . The significance of this research has been underscored by recent attention to this subject. Our approach involved ARPES measurements to examine the electronic contribution and Raman scattering measurements to evaluate the contribution from the lattice degree of freedom. Our thorough

analysis of ARPES and Raman data points toward electron-phonon coupling as the likely the primary driving force behind the CDW in ScV₆Sn₆.

In our earlier work on GdV₆Sn₆ [*Sci. Adv.* 8, eadd2024 (2022)], the focus was on identifying the topologically nontrivial Dirac surface states (TDSSs) around the Brillouin zone (BZ) center (Γ point) and van Hove singularities (VHSs) around the BZ boundary (M point), along with their tunabilities.

Sharing the same crystal structure with its sister compound GdV₆Sn₆, ScV₆Sn₆ exhibits certain similarities in electronic structure, including the existence of TDSSs around the Γ point and VHSs around the M point. As outlined in the introduction section of our manuscript, ScV₆Sn₆ undergoes a unique three-dimensional CDW phase transition with a wave-vector $\mathbf{Q} = (1/3, 1/3, 1/3)$, in sharp contrast to the non-CDW compound GdV₆Sn₆ and the $2 \times 2 \times 2$ CDW kagome metals AV₃Sb₅ ($A=K, Rb, Cs$). Notably, our ARPES measurements on ScV₆Sn₆ reveal an intriguing VHS near the \bar{K} point, a feature distinctly absent in GdV₆Sn₆.

2、 The authors indicated that the VHSs near the K point could introduce nesting wave vector that aligning with the observed CDW wave vector, and it suggests that electronic correlation might play a role in promoting the in-plane component of the CDW order. However, they do not directly address how a nesting wave vector from a VHS far away from the Fermi level contributes to the CDW. The authors said it could potentially play a role in promoting the in-plane component of CDW order, particularly given the noticeable coupling between V and Sc/Sn atoms. I think a more detailed theoretical explanation might be necessary if the authors insist in this point in the main text.

Fig. R2. Susceptibilities as a function of temperature in the particle-hole channel with the momentum transfer \mathbf{Q}_1 for the VHSs around K point. **a Temperature-dependent susceptibilities for different effective mass ratios. **b** Same data as in (a), but for different chemical potentials with $m_y/m_x=5$.**

When VHSs are situated in close proximity to the Fermi level (E_F), Fermi surface nesting occurs, leading to a significant susceptibility in the particle-hole channel at low temperatures (see Fig. R2a). In the presence of interactions, the charge susceptibility can diverge, promoting a CDW instability. As VHSs move away from E_F , the corresponding charge fluctuation weakens but remains noteworthy (temperature-dependent susceptibilities for different chemical potential, as depicted in Fig. R2b). It is essential to note that pure electronic interactions alone cannot induce a CDW instability.

In the case of ScV₆Sn₆, a distinctive feature is the strong electron-phonon coupling and phonon softening. The $\sqrt{3} \times \sqrt{3}$ $R30^\circ$ in-plane lattice instability related to Sc/Sn atoms can be further enhanced by the substantial charge fluctuations from the V-d bands and the coupling between V and Sc/Sn atoms. Our interpretation is that the primary driving force behind the CDW in ScV₆Sn₆ stems

from the lattice degree of freedom, with VHSs providing a secondary contribution, even when they are not in close proximity to E_F .

To further illustrate the potential role of the VHS around the K point, we have included the theoretical calculations (Fig. R2) in Fig. S4 of the revised Supplementary Materials.

3. The temperature-dependent ARPES measurements show negligible differences between the Kagome bands in the CDW phase and normal state. While acknowledging the potential significance of negative findings in scientific investigations, I am uncertain whether these results possess sufficient importance to warrant publication in high-impact journals such as Nature Communications since several previous ARPES works have reported the similar result on that. Consequently, I would like to defer this question to the editor.

The unique CDW in ScV_6Sn_6 has garnered significant recent attention in the scientific community. Although a few related preprints appeared slightly earlier than our work, **notably, our study provided the first systematic and scientific ARPES measurements on ScV_6Sn_6** (*arXiv*: 2304.06431). Our high-resolution ARPES measurements unveiled multi-orbital characteristics and a relatively apparent three-dimensionality in the electronic structure of ScV_6Sn_6 . Moreover, distinct electronic states, including the TDSSs, Dirac cones and multiple VHSs in the vicinity of E_F , with one VHS near the \bar{K} point exhibiting nesting wave vectors in proximity to the $\sqrt{3} \times \sqrt{3}$ $R30^\circ$ CDW wave vector, were identified.

We respectfully disagree with the Reviewer's characterization of our findings as "negative". Instead, **our ARPES results, combined with our Raman measurements, provide important insights into the fascinating CDW observed in the kagome metal ScV_6Sn_6 .** Our systematic ARPES measurements suggest a weak band reconstruction and folding effect in ScV_6Sn_6 , indicating a small distortion of V atoms. These findings imply that Fermi surface nesting may not be the dominant driving force behind the CDW in ScV_6Sn_6 . Our Raman measurements reveal a two-phonon mode in the normal state and new emergent Raman-active phonon modes in the CDW phase, indicating strong electron-phonon coupling. Thus, the combined insights from the ARPES and Raman measurements underscore the dominant role of lattice degrees of freedom in promoting the CDW in ScV_6Sn_6 .

For the information, I noted that some latest ARPES studies on ScV_6Sn_6 revealed energy gaps at the Fermi level [Arxiv 1304.11820; Commun Mater 4, 103 (2023)]. These results appear to diverge from those reported by the authors.

We have taken note of the two ARPES studies [*arxiv*: 2304.11820; *Commun. Mater.* **4**, 103 (2023)] on ScV_6Sn_6 . However, based on our systematic ARPES measurements, we find the claimed energy gap has alternative interpretations.

1) The suggested energy gap in the preprint (*arxiv*: 2304.11820) appears to be present in the normal state (120 K, as highlighted by the orange arrow in Fig. R3a), indicating its irrelevance to the CDW. Moreover, we have observed a similar "energy gap" feature on the β band in our ARPES data (Fig. R3c-e). However, the β band contributing to the energy gap in the preprint is much weaker in the ARPES spectrum collected along a different $\bar{\Gamma} - \bar{M}$ direction (Cut #2 in Fig. R3c and e), and it appears to be well reproduced by the β band in the $k_z = 0$ plane [Fig. R3f(i)], while the experimental α band in Cut #1 (Fig. R3c and d) is consistent with the theoretical α band in the $k_z = \pi$ plane [Fig. R3f(ii)]. Therefore, we do not consider the appearance of the β band as evidence for the CDW energy gap. Instead, it would be more reasonable to attribute the coexistence of the α band β bands (Fig. R3a, d and e) to k_z broadening effects.

2) The ARPES data in the work by Manuel Tuniz *et al.* [*Commun. Mater.* **4**, 103 (2023)] involve band structures from different surface terminations, as a large beam spot was used. Consequently, the suggested energy gap in their work is also likely due to sample position vibrations during their temperature-dependent ARPES measurements. The [Peer Review File](https://doi.org/10.1038/s43246-023-00430-y) (<https://doi.org/10.1038/s43246-023-00430-y>), published together with the paper, has raised these concerns about their ARPES data.

We would like to emphasize that whether the above-discussed energy gap exists or not, it does not affect our main conclusion that there is a weak band reconstruction and folding effect on V-*d* orbital bands. Even if the potential energy gap is present in ScV₆Sn₆, it significantly differs with the dramatic band reconstructions on the V bands observed in CsV₃Sb₅ [*npj Quantum Mater.* **8**, 67 (2023)]. Thus, in any case, our ARPES measurements suggest that Fermi surface nesting may not be the dominant driving force behind the CDW in ScV₆Sn₆.

Fig. R3. Potential band reconstruction in ScV₆Sn₆. **a,b** ARPES spectra measured along the $\bar{\Gamma} - \bar{M}$ direction at 120 K (a) and 6 K (b), adapted from the preprint (*arXiv*: 2304.11820). **c-d** ARPES spectra taken along two different $\bar{\Gamma} - \bar{M}$ directions, *i.e.*, Cut #1 (d) and Cut #2 (e); the momentum paths are indicated by the red line in (c). The differences in the two cuts are likely due to the matrix element effects. The data were taken at 20 K using 88 eV photons corresponding to the $k_z = \pi$ plane. **f** Orbital character resolved electronic structure for the V *d* and Sn *p* orbitals in the normal state of ScV₆Sn₆ from the $k_z = 0$ (i) and $k_z = \pi$ (ii) planes, respectively.

Summary of Modifications

In the revised manuscript, the modifications are highlighted in red font.

1. In light of Reviewer #2's comments, we have tempered our language concerning the assignment of the A_3 mode in the revised manuscript, allowing for alternative interpretations.

Furthermore, in Section 6 of the Supplementary Materials, we have added extra sentences to explore potential scenarios that could elucidate the emergence of new modes at low temperatures.

2. Following the suggestion of Reviewer #3, we have included additional theoretical calculations in Fig. S4 of the revised Supplementary Materials to further illustrate the potential role of the VHS around the K point.